# Beyond Greedy Exits: Improved Early Exit Decisions for Risk Control and Reliability

**Divya Jyoti Bajpai & Manjesh Kumar Hanawal**
Department of Industrial Engineering and Operations Research
Indian Institute of Technology Bombay
Powai, Maharashtra, India
{divyajyoti.bajpai, mhanawal}@iitb.ac.in

## Abstract

Early-Exit Deep Neural Networks enable adaptive inference by allowing prediction at intermediary layers, significantly reducing computational costs and latency. Most of the early exit strategies greedily exit a sample at an intermediary layer if the confidence in class prediction exceeds a predefined threshold that is set using a static validation set. This is problematic as the model might be overconfident in a wrong class. Also, they are not robust to distribution shifts encountered in deployment, which can undermine model trustworthiness and accuracy. To address these challenges, we propose UAT that adapts the threshold for exit decisions using a Multi-Armed Bandit framework, enabling online, unsupervised adjustment of exit decisions. UAT makes decisions based on a new reward function that assesses predictive certainty and its reliability to balance computational efficiency and prediction quality while penalizing unnecessary late exits. We provide guarantees on risk achieved by UAT and validate its performance on diverse tasks spanning vision-language understanding, text generation, and classification. Our framework demonstrates consistent improvements in speedup $(1.70 - 2.10\times)$ with a minimal performance drop $(< 2\%)$ as compared to full model performance. Our source code is available at https://github.com/Div290/UAT.

## 1 Introduction

As Deep Neural Networks (DNNs) continue to scale in size and complexity, the cost of inference has become a major bottleneck—manifesting as increased latency, higher energy consumption, and a growing carbon footprint [37, 39]. These challenges highlight the need for efficient and sustainable deployment strategies for large models. Moreover, real-world environments often impose dynamic computational constraints, driven by resource scarcity or energy availability [40]. To remain practical and responsive, modern DNNs must adapt computational load in real time and consume resources as much as necessary, as per the complexity of the samples.

Early-Exit DNNs (EEDNNs) [47] offer a compelling solution for adaptive inference under dynamic computational constraints. EEDNNs can produce intermediate predictions at various depths of the DNN, allowing the network to 'exit' the sample early without passing it through all the layers. This mechanism enables dynamic resource utilization and significantly reduces computation and inference latency. This makes EEDNNs well-suited for deployment under resource constraints across diverse domains, including computer vision [47, 4, 29], natural language processing [12, 9], and vision-language systems [46]. However, this flexibility comes at a cost: initial layers are more prone to making riskier predictions, which degrade the trust in the model [36]. Consequently, EEDNNs face a fundamental trade-off—balancing efficiency gains against potential degradation in predictive accuracy [30]. This trade-off is governed by the exit decisions that are crucial in EEDNNs.

39th Conference on Neural Information Processing Systems (NeurIPS 2025).

In existing methods, the decision to exit depends on two variables: confidence and threshold. Confidence is usually defined as the entropy of the prediction [50], prediction consistency [52], or ensemble methods [8, 49]. Threshold is a value against which confidence is compared. If the confidence value exceeds the threshold, the model is considered confident enough, and the sample is allowed to exit without further processing (early exit). However, there are some crucial concerns in existing methods: 1) The confidence is not a reliable metric as the model may get overly confident at the initial layers on a wrong class, making the sample exit with an incorrect label [36]. Fig. 1 discussed in the Appendix 5.1 shows this phenomenon, where a large number of samples can exit from the intermediary layer with high confidence on the wrong class. 2) The threshold to make an exit decision is precomputed using a small validation set, assuming it generalizes well to the inference dataset distribution. But this assumption does not hold in real-world conditions, where distribution shifts are common. This defeats the goal of low-risk and reliable inference and necessitates the need for an improved strategy that incorporates the reliability of model output in the confidence metric and dynamically adjusts thresholds as per the input data distributions.

During inference, the data arrives in an online fashion. Hence, the exit decisions need to be adapted in an online and unsupervised manner (a quality also critical for zero-shot tasks). To address these challenges, we introduce a novel framework for *dynamic threshold adaptation* in EEDNNs, aimed at optimizing the trade-off between prediction performance and computational costs. In contrast to existing methods that rely on confidence at an intermediary layer exceeding a fixed threshold, our early exit strategy uses a *confidence metric* that accounts not only for the model's predictive certainty but also the *reliability* of that certainty.

We derive an exit strategy using the Multi-Armed Bandit (MAB) [3] framework, where each arm represents a candidate exit threshold. The bandit agent aims to learn a threshold that optimizes a custom reward that involves the confidence metric and a penalty for late exits to discourage unnecessary computation. We theoretically prove that optimizing this metric aligns with maximizing the true class probability, providing a principled foundation for early-exit decisions.

Our framework promotes efficient inference without compromising performance, and we demonstrate its effectiveness across diverse tasks, including vision-language understanding (e.g., image captioning [34], VQA [20]), text generation (e.g., summarization [25], QA [38]), and classification (e.g., NLI, sentiment analysis [48]). In summary, our contributions are as follows:

- We introduce the notion of reliability scores on the confidence of the early exit output to improve the trustworthiness of EEDNNs.

- We propose a dynamic early exit framework that adapts to distributional changes at test time, enhancing both *efficiency* and *robustness* in EEDNNs.

- We apply the bandit framework to learn the optimal strategy. We develop an upper confidence-based algorithm and establish a theoretical bound on the risk achieved by it.

- We validate our approach on a broad set of tasks, demonstrating consistent gains in both computational efficiency and predictive accuracy.

## 2 Related works

Early Exit methods have been applied to various tasks such as image classification, text classification, text generation and vision language tasks to reduce the computational latency and inference latency. Below, we discuss some of the works on risk control in EEDNNs.

**Early Exits:** BranchyNet [47] first proposed the early exit method for image classification tasks. It uses classification entropy as a confidence metric. Shallow-deep [32] and MSDNet [29] improve upon BranchyNet by effectively choosing the thresholds based on the confidence threshold. ZTW [49], JEI-DNN [18] and BEEM [8] propose different methods for assessing the confidence. ZTW uses multiple classifiers to decide exiting, JEI-DNN learns a gating function during training to decide exiting, while BEEM builds a confidence measure using ensemble methods. DeeCAP [24], MuE [46], CapEEN[5] and FREE [10] implement early exits to vision language tasks.

For the Language tasks, and specifically in Large Language Models (LLMs), the EE methods have been popular [11, 15, 35, 13, 45, 21, 27, 14, 44, 31]. DeeBERT [50] first applied it to the BERT model. Later, PABEE [52] developed a patience-based method to decide exiting. BERxiT [51] proposes an

efficient fine-tuning strategy for EE-based models. LeeBERT [53] performs self-distillation between exits to effectively share the knowledge across layers with exits. ETFEE [31] adds an adapter on top of the transformer layer and an entangled frame classifier to make exits learn better. CeeBERT [6] and DAdEE [7] provide methods to adapt the EEDNN to various domains. CALM [41] extends the idea of EEs to language generation tasks.

**Risk in EEDNNs:** As EEDNNs consist of multiple exit classifiers, the initial layers of the EEDNNs with low-level features might leave the sample at risk. EERC [30] shows that the threshold, if chosen properly, can also minimise the risk of EEDNNs. MIE-LAP [36] addresses the overconfidence in EEDNNs by uncertainty quantification. It proposes an approach for uncertainty-aware decision-making by leveraging the last layer Laplace approximation implementation.

While existing early-exit approaches determine a fixed confidence threshold using a validation set and apply it uniformly during inference, this static strategy often permits overconfident yet incorrect predictions to exit prematurely. In contrast, our method introduces three key innovations: **1) Dynamic adaptation to test-time distribution:** We tailor the exit decisions based on the distributional characteristics of the test data. **2) Confidence calibrated by reliability:** Instead of maximizing the raw confidence scores, we explicitly account for their reliability, leading to more robust exit decisions. **3) System-level risk modeling:** Rather than comparing early exits against the final layer's predictions, we model the overall risk of the complete system, enabling our method to potentially surpass the performance of the base DNN itself, an important limitation overlooked by prior work.

# 3 Preliminaries

Consider the input-output space denoted as $\mathcal{X} \times \mathcal{Y}$. A sample, denoted $(x, y)$, is drawn from an underlying probability distribution $\mathcal{P}$. For classification tasks, the label space is given by $\mathcal{Y} = \{1, 2, \ldots, K\}$, whereas for regression, the output set is a subset of $\mathbb{R}^d$.

## 3.1 The Notion of Risk

In statistical modeling, risk control mechanisms [2, 16] enhance the reliability of predictive models by imposing constraints on threshold-based decision processes. Let $\Omega$ denote a finite set of thresholds. Consider a model $f_\tau$ parameterized by a threshold $\tau \in \Omega$, which outputs a set of predictions based on confidence levels. For a classification task, the set-valued predictor $f_\tau : \mathcal{X} \to 2^{\mathcal{Y}}$ is defined as $f_\tau(x) = \{\hat{y} \in \mathcal{Y} \mid p(\hat{y} \mid x) \geq \tau\}$, where $p(\hat{y} \mid x)$ is the probability that model assigns label $\hat{y}$ to $x$. If no class probability exceeds $\tau$, the predictor abstains by returning an empty set ($f_\tau(x) = \emptyset$).

To assess reliability, we define a miscoverage loss that penalizes the exclusion of the true label as $\ell(f_\tau(x), y) = \mathbb{1}[y \notin f_\tau(x)]$, where $\mathbb{1}[\cdot]$ is the indicator function. The associated risk is the expected probability of miscoverage:

$$\mathcal{R}(\tau) = \mathbb{E}_{(x,y)\sim\mathcal{P}}[\ell(f_\tau(x), y)] = \mathbb{E}_{(x,y)\sim\mathcal{P}}[\Pr\{y \notin f_\tau(x)\}].$$

For a given tolerance, $\epsilon \in (0, 1)$ and $\delta \in (0, 1)$, we say that the threshold set $\hat{\Omega}$ controls the risk with within tolerance $\epsilon$ with probability at least $1 - \delta$ if

$$\mathbb{P}(\mathcal{R}(\tau) \leq \epsilon) \geq 1 - \delta, \forall \tau \in \hat{\Omega}. \tag{1}$$

The advantages of risk control frameworks are significant: (i) they require no assumptions on the data distribution $\mathcal{P}$, making them distribution-free and (ii) they can be applied post-hoc to any model. These features make them particularly valuable for practical deployment in uncertain environments.

## 3.2 Early Exit Mechanisms in Deep Neural Networks

Early Exit Deep Neural Networks (EEDNNs) extend the architecture of the DNNs by attaching Exit Classifiers ($ECs$) at the intermediate layers. The EC at layer intermediate layer $i$ produces a probability distribution over $\mathcal{Y}$ denoted by $p_i(\cdot|x) = p(\cdot|x, \theta_i) = EC_i(h_i)$ where $h_i = \phi_i(x)$ is the hidden representation output by the $i$th layer of the model with $i \in \{1, 2, \ldots, L\}$ and $\theta_i$ is the set of parameters consisting of the EEDNN parameters till layer $i$ comprising of both $EC_i$ and main model parameters. The final prediction layer, indexed by $L$, corresponds to the standard DNN where all layers are utilized for inference. Each $EC_i$ estimates the class label as $\hat{y}_i = \arg\max_{y \in \mathcal{Y}} \hat{p}_i(y|x)$

with a confidence score $C_i := C_i(x) = \max_{y \in \mathcal{Y}} p_i(y|x)$. $C_i$ measures the certainty associated with the prediction. However, this confidence might not be very reliable as it provides confidence in the predicted class, which might be wrong due to overconfident EEDNNs.

During inference, the EEDNN exits from a layer where the confidence of $EC$ exceeds the predefined threshold $\tau_i \in [0, 1]$ for the first time. Following previous works [50, 30], a single global threshold is considered across all the $ECs$, i.e., $\tau_i = \tau$ for every $i \in \{1, 2, \ldots, L\}$. Then, for a given $\tau$, the probability that $x$ exits with a label $\hat{y}$ is given by

$$p_\tau(\hat{y} \mid x) = \begin{cases} p_L(\hat{y} \mid x), & \text{if } C_k < \tau \ \forall \, k \in \{1, 2, \ldots, L-1\} \\ p_i(\hat{y} \mid x) \text{ where } i = \min \{k \in \{1, \ldots, L-1\} \mid C_k \geq \tau\}, \text{ otherwise} \end{cases} \tag{2}$$

As the class label assigned by the EEDNNs relies on the confidence score, their reliability is important for trustworthy predictions. The threshold $\tau$ directly influences the balance between computational cost and model accuracy. Lower values encourage premature exits, reducing computational overhead but potentially sacrificing performance.

The risk metric defined in the previous subsection can be naturally applied to EEDNNs. As samples exit if the confidence score exceeds a given threshold, we can associate risk for each threshold and aim to identify the threshold that yields the least risk. However, the task of the threshold in early exits is not only limited to minimizing the risk but also reducing the inference latency. Hence, a threshold that is the smallest from the set $\hat{\Omega}$ is desired. That is, instead of looking for a set $\hat{\Omega}$ that satisfies Eqn. 1, we aim to identify threshold $\tau^* = \min\{\tau \in \hat{\Omega}\}$. The proportion of samples that exit earlier is higher for $\tau^*$ than any other $\tau$ in $\hat{\Omega}$.

For a given threshold $\tau$, let $p_\tau(y \neq \hat{y}|x)$ denote the probability that the EEDNNs makes errors on sample $(x, y)$. Then the risk of the EEDNNs is defined as

$$\mathcal{R}(\tau) = \mathbb{E}_{(x,y) \sim \mathcal{P}} \left[ p_\tau(y \neq \hat{y}|x) \right].$$

Our objective is to find a threshold $\tau$ for which $\mathcal{R}(\tau)$ lies within a given tolerance value with high probability. The value of $\tau$ that meets the requirement is a function of the sample distribution $\mathcal{P}$, and the distribution seen in real deployment could drift from that of the training samples used to train the EEDNNs. The best value of $\tau$ depends on the distribution of the input sample, and hence needs to adapt to any drift in the sample distributions. This adaptation requires knowledge of risk. However, this information cannot be estimated during the inference time as ground truth labels are not available. To address this challenge, we perform the learning in two stages.

In the first stage, we perform offline training of the EEDNN and a linear layer network that will allow us to express the risk without knowing the ground truth labels. This training involves estimating the reliability of the confidence scores output by the exit classifiers. In the second phase, we use this proxy for risk and learn the best threshold $\tau$ by maximizing a reward function using the Multi-Armed Bandit (MAB) framework. The following Lemma gives a simple relation that we use to obtain a proxy for a risk function.

**Lemma 3.1.** *Given a threshold $\tau$, let $\hat{y}$ be the label predicted by EEDNNs on sample $(x, y)$. Define $p_\tau(y = \hat{y}|\hat{y}, x)$ be the probability that the predicted label is the true label. Then $p_\tau(y = \hat{y}|x) = p_\tau(\hat{y}|x) \cdot p_\tau(y = \hat{y}|x, \hat{y})$*

The proof is straightforward and is given in the Appendix A.1. $p_\tau(y = \hat{y}|x, \hat{y})$ is the probability that the model's prediction is correct. Note that $p_\tau(\hat{y}|x)$ can be estimated during inference as it does not depend on the label. Then, the lemma suggests that we can compute risk by estimating $p_\tau(y = \hat{y}|x, \hat{y})$. The next subsection discusses training a neural network to estimate it in an offline fashion.

### 3.3 Exit and risk function training

We train a reliability function (neural network) $g$ that assigns a reliability score to the class probabilities output by the exit classifiers. We use a modified loss function that is used to train the exit classifiers in EEDNNs to train $g$. In the conventional EEDNNs, the objective at each exit is to maximize the probability of the correct label using cross-entropy loss. We augment this objective with the reliability function $g$. For a sample $(x, y) \sim \mathcal{P}$, we define a modified loss is as:

$$\mathcal{L}_i = \mathcal{L}_{\mathrm{CE}}(p_i(\cdot|x), y)(1 + g(p_i(\cdot|x), i) + \Phi(c - \phi(g)). \tag{3}$$

The $\mathcal{L}_{\mathrm{CE}}(p_i(\cdot|x), y)$ in the first term is the standard cross-entropy loss that encourages the model to correctly classify the input. It is multiplied by a factor $1 + g(\cdot)$, to link the sample-wise loss with the reliability function $g$, which estimates the confidence of the model's prediction. The minimization of $\mathcal{L}_i$ causes $g$ to assign high values to samples with low cross-entropy (i.e., correctly and confidently classified), and low values to those with high loss (i.e., ambiguous or misclassified samples). Intuitively, this creates a competitive dynamic where $g$ is penalized for activating on high-loss (uncertain) samples and is incentivized to concentrate its high values on low-loss samples.

When $g(\cdot) \geq 0.5$, it indicates that the model output is reliable. $\phi(g)$ in the second term denotes the fraction of samples for which the model's output is reliable. $\Phi(c - \phi(g))$ acts as a regularizer to enforce that the model is reliable for at least $c$ fraction of samples. The function $\Phi(a) = \max(0, a)^2$ penalizes the model when the coverage $\phi(g)$ falls short of the desired threshold $c_i$. This regularizer restricts $g$ from giving a trivial solution $g \equiv 0$, as such a solution would provide no meaningful discrimination between easy and hard samples. The role of $\Phi(c - \phi(g))$ is to prevent this collapse by ensuring that the model is reliable on at least $c$-fraction of the training samples. $c$ can be a function of the exit layer that can be chosen based on validation dataset (see Appendix B.2). We empirically observe that our loss does not impact the model's predictive performance (see Appendix B.9).

Once trained, $g$ can be used as a proxy for the model's correctness probability $p(y = \hat{y} \mid x, \hat{y})$. To account for the cost associated with deeper exits [32], the final loss across all exits is computed as a weighted average $\mathcal{L} = \frac{\sum_{i=1}^{L} i \cdot \mathcal{L}_i}{\sum_{i=1}^{L} i}$, which assigns higher weights to exits at deeper layers where inference cost is higher.

### 3.4 Dynamic thresholding during inference

We first describe the MAB setup to dynamically adapt to the threshold during inference. In a MAB setup, a decision-maker iteratively selects actions, adapting to an unknown environment. The goal is to identify an action that gives the highest reward. In our setup, the actions are the set of thresholds.

We next define the reward for each threshold. For a given threshold/action $\tau$, let $C_\tau^i$ denote the confidence model has in the predicted class when the sample exits from $EC_i$. We define $C_g^i = 1 - g(p_i(\cdot|x))$ as a score of underlying risk (unreliable) in the confidence score of the model at $EC_i$. We set $1 - C_g^i$ as an approximation of $p_\tau(y = \hat{y}|x, \hat{y})$. Then Lemma 3.1 indicates that the $p_\tau(y = \hat{y}|x)$ can be well approximated as $C_\tau^i(1 - C_g^i)$ when sample $x$ exits from $EC_i$. However, a threshold that maximizes $C_\tau^i \cdot (1 - C_g^i)$ may result in suboptimal efficiency gains as it will penalize the thresholds that are forcing the sample to move to deeper layers. To account for this fact, we define reward for a threshold $\tau \in \Omega$ on a sample as follows:

$$r(\tau) = \begin{cases} C_\tau^i \cdot (1 - C_g^i) - \psi(i), & \text{if } (C_\tau^i \cdot (1 - C_g^i) \geq \tau) \cap (i < L) \\ C_\tau^i \cdot (1 - C_g^i) - \psi(L), & \text{if } i = L \end{cases} \tag{4}$$

where $\psi(\cdot)$ is a penalty to disincentivize the sample from moving to a deeper layer unnecessarily. In our setup, we define $\psi(i) = \lambda \cdot i$ where $\lambda$ could be interpreted as the processing cost per layer.

The expected reward for arm $\tau \in \Omega$ is $\mathbb{E}[r(\tau)] = \sum_{i=1}^{L} \mathbb{E}[C_\tau^i(1 - C_g^i) - \lambda \cdot i]P(i)$ where $P(i)$ is the probability that the sample exits from $i$th layer and expectation is with respect to the randomness of confidence and reliability scores. Let $\tau^* = \arg\max_{\tau \in \Omega} \mathbb{E}[r(\tau)]$ denote the optimal threshold for a given $\lambda$. Now consider a policy $\pi$ that selects the threshold $\tau_t \in \Omega$ based on past observations. For a given number of rounds, the performance of policy $\pi$ is evaluated using the expected cumulative regret defined as $R(\pi, T) = \sum_{t=1}^{T} \mathbb{E}(r(\tau^*) - r(\tau_t))$, where the expectation is with respect to randomness in the selection of thresholds induced by the past sample. A policy $\pi$ that satisfies $R(\pi, T)/T \to 0$ is said to be sub-linear and plays most of the time the optimal threshold. Our goal is to develop an algorithm that is sub-linear and has a small linear component. As we will show later, this translates to achieving a small empirical risk defined as $\hat{\mathcal{R}}(\pi) = 1 - \frac{\sum_{t=1}^{T} p_{\tau_t}(y_t = \hat{y}|x_t, \hat{y}_t)}{T}$ where $(x_t, y_t)$ is the input sample, $\hat{y}_t$ is the predicted label, and $\tau_t$ is the threshold selection by $\pi$ in round $t$.

## 4 Algorithm

We develop an algorithm named UAT. Its pseudo-code is given in 1. The inputs to the algorithm are the exploration constant $\gamma$ and the penalizing term $\psi(i)$. For the first $|\Omega|$ samples, the algorithm plays each arm once. In the subsequent rounds, it plays arm with the highest Upper Confidence Bound (UCB) index denoted as $\tau_t$. UCB indices are obtained by taking the weighted sum of the empirical average rewards and the confidence bonuses with $\gamma$ as the weight factor. If $C_\tau^i(1 - C_g^i)$ at the $i$th layer is greater than $\tau_t$, then the sample exits; otherwise, the sample is passed to the next layer in the backbone. If the sample does not exit at any intermediate classifier, then it is inferred at the final layer. Finally, the algorithm updates the number of pulls $(N(\tau_t))$ and the empirical mean $(Q(\tau_t))$ of the played arm. Note that the algorithm is applied in the inference phase. In Table 6, we empirically show how different components contribute to the reward function and discuss in Appendix B.1. We also show in Appendix B.8 that UAT has negligible computational cost. The following result establishes the average risk achieved by UAT.

**Theorem 4.1.** *Let $(1 - C_g^i)$ approximates the value of $p_\tau(\hat{y} = y|x, \hat{y})$ with probability $(1 - \delta_1)$. Let the risk associated with $\tau^*$ be bounded by $\epsilon^*$. Then, for sufficiently large $T$ and given tolerance $\epsilon$, UAT achieves $\mathbb{P}(\hat{\mathcal{R}}(\pi) \le \epsilon^d) \ge (1 - \delta_1)(1 - \delta')$. where $\epsilon^d = \epsilon + \epsilon^*$, $\lambda \le \frac{\epsilon}{L}$ and $\delta'$ is a constant.*

The detailed proof is given in the appendix. Note from the theorem that $\epsilon^*$ is equivalent to the full model risk (that is always there), and the additional risk of EEDNN due to early exits is $\epsilon$. Then the risk in EEDNN performance is $\epsilon^d = \epsilon + \epsilon^*$. This theorem also suggests that a lower value of $\lambda$ can lead to lower risks, which is true based on our reward function. If $\lambda$ is chosen small, then more emphasis is given to the $C_\tau^i(1 - C_g^i)$ value which maximizes $p(y = \hat{y}|x, \hat{y})$. The proof also provides us a way to fix the $\lambda$ value, as we want the maximum $\lambda$ value such that the risk levels are below $\epsilon$. Hence we choose $\lambda = \frac{\epsilon}{L}$ in this work. This theorem bounds the risk of the EEDNN where the threshold to decide exiting is chosen based on the policy $\pi$ given by algorithm 1. Also, observe from the proof outline that it suggests that the risk of the EEDNN will be greater than $\epsilon^*$, which denotes the optimal risk. The proof steps also provide the link of risk with the regret.

## 5 Experiments

In this section, we provide empirical results of our work on various tasks such as text classification, language modelling and vision language tasks.

---

**Algorithm 1** UCB-based Adaptive Thresholds (UAT)

---

1: **Input:** $\psi(i), \gamma \ge 1$
2: **Initialize:** Play each threshold once. Observe $r(\tau)$ and set $Q(\tau) \leftarrow \mathbf{0}, N(\tau) \leftarrow \mathbf{1}, \forall \tau \in \Omega$.
3: **for** $t = |\Omega| + 1, |\Omega| + 2, \cdots$ **do**
4: $\quad$ Observe an instance $x_t$
5: $\quad \tau_t \leftarrow \arg\max_{\tau \in \Omega} \left( Q(\tau) + \gamma\sqrt{\frac{\ln(t)}{N(\tau)}} \right)$
6: $\quad$ **for** $i = 1$ **to** $L$ **do**
7: $\quad\quad$ Pass $x_t$ till layer $i$ and apply threshold $\tau_t$ and observe $S_{\tau_t} = C_{\tau_t}^i \cdot (1 - C_g^i)$
8: $\quad\quad$ **if** $S_{\tau_t}^i \ge \tau_t$ and $i < L$ **then**
9: $\quad\quad\quad$ Infer at layer $i$ and exit
10: $\quad\quad\quad r_t(\tau_t) \leftarrow S_{\tau_t}^i - \psi(i), N_t(\tau_t) \leftarrow N_{t-1}(\tau_t) + 1, Q_t(\tau_t) \leftarrow \frac{\sum_{j=1}^t r_j(\tau_j)\mathbb{1}_{\{\tau_j = \tau_t\}}}{N_t(\tau_t)}$
11: $\quad\quad\quad$ **break**
12: $\quad\quad$ **else if** $i = L$ **then**
13: $\quad\quad\quad$ Process and infer at the last layer.
14: $\quad\quad\quad r_t(\tau_t) \leftarrow S_{\tau_t}^L - \psi(L)), N_t(\tau_t) \leftarrow N_{t-1}(\tau_t) + 1, Q_t(\tau_t) \leftarrow \frac{\sum_{j=1}^t r_j(\tau_j)\mathbb{1}_{\{\tau_j = \tau_t\}}}{N_t(\tau_t)}$
15: $\quad\quad$ **end if**
16: $\quad$ **end for**
17: **end for**

---

| Model/Datasets | SST-2 | MNLI | RTE | QNLI | QQP | SQuAD | Speedup |
|---|---|---|---|---|---|---|---|
| Full model | 95.2 | 86.0 | 67.7 | 92.0 | 72.5 | 83.0 | 1.00× |
| *EE models* | | | | | | | |
| PABEE | 94.8 | 85.0 | 67.2 | 91.1 | 72.3 | 82.0 | 1.53× |
| ZTW | 94.5 | 85.1 | 66.9 | 90.6 | 72.0 | 81.7 | 1.70× |
| MuE | 94.9 | 85.5 | 67.3 | 91.3 | 72.3 | 82.2 | 1.59× |
| JEI-DNN | 95.1 | 85.6 | 67.5 | 91.5 | 72.2 | 82.3 | 1.49× |
| BEEM | 95.4 | 86.1 | 67.6 | 91.9 | 72.4 | 82.4 | 1.79× |
| *Risk-based methods on EE models* | | | | | | | |
| CALM | 95.5 | 85.9 | 67.7 | 91.6 | 72.3 | 82.2 | 1.39× |
| MIE LAP | 95.3 | 85.8 | 67.5 | 91.7 | 72.1 | 81.8 | 1.45× |
| EERC | 95.0 | 85.6 | 67.3 | 91.5 | 71.9 | 81.6 | 1.50× |
| **Ours** | **95.8** | **86.6** | **68.3** | **92.4** | **73.0** | **83.2** | **1.85×** |

Table 1: Results on the BERT-large model showcasing accuracy and average speedup (Speedup).

| Model | Summarization | | | | QA | MT | Speedup |
|---|---|---|---|---|---|---|---|
| | SamSum | CNN | Multi-News | BIGPAT. | SQuAD | IWSLT | |
| Full model | 48.82 | 41.15 | 37.62 | 49.68 | 90.63 | 39.19 | 1.00× |
| *EE-based methods* | | | | | | | |
| PABEE | 42.25 | 37.54 | 32.97 | 43.71 | 87.39 | 35.80 | 1.90× |
| ZTW | 43.96 | 39.07 | 33.62 | 45.38 | 88.02 | 36.75 | 1.83× |
| MuE | 43.48 | 38.11 | 33.73 | 45.02 | 87.53 | 36.32 | 1.82× |
| JEI-DNN | 44.45 | 37.78 | 34.52 | 45.39 | 88.14 | 36.94 | 2.05× |
| BEEM | 45.19 | 39.82 | 35.07 | 47.43 | 88.28 | 37.06 | 1.97× |
| *Risk-based EE methods* | | | | | | | |
| CALM | 46.73 | 40.29 | 36.14 | 48.02 | 88.79 | 37.84 | 1.70× |
| MIP LAP | 44.87 | 39.58 | 35.92 | 47.16 | 88.47 | 37.91 | 1.76× |
| EERS | 46.79 | 40.52 | 36.83 | 47.96 | 89.02 | 37.68 | 1.63× |
| **Ours** | **48.09** | **40.97** | **37.10** | **48.74** | **90.59** | **38.62** | **2.10×** |

Table 2: Results on the T5-large model showcasing task metrics with average speedup (Speedup).

**Datasets:** For text classification tasks, we use the standard GLUE [48] datasets that consist of three types of classification tasks: 1) Sentiment analysis (SST-2), 2) Text entailment (RTE, QQP) and 3) Natural Language Inference (MNLI, QNLI). For Question-Answering tasks, we use the SQuAD 2.0 [38] dataset. For text summarization tasks, we use a diverse set of datasets covering different domains and styles: 1) Dialogue summarization (SAMSum) [25], 2) Long-form technical summarization (BIGPATENT) [43], 3) Multi-document summarization (MultiNews) [23], and 4) News summarization (CNN/DailyMail) [28]. For machine translation tasks, we use the IWSLT De-En dataset [17], which consists of translated transcripts from German to English. It serves as a standard benchmark for evaluating translation quality on spoken language content. For vision language tasks, we use the COCO [34] and NoCaps [1] datasets for Image captioning, VQAv2 [26] dataset for visual question answering and VisDial [20] for visual dialogue.

**Metrics:** We report key metrics such as accuracy for text classification, F1 score for Question answering tasks, Rouge-L score for text summarization, BLEU-4 (B4), CiDER (C), Spice (S) and Meteor (M) for image captioning. VQA accuracy for VQA tasks and Mean Reciprocal Rank (MRR) for visual dialogue. Following existing baselines [22, 46], speedup is calculated as the acceleration of average inference time per token compared to the full model. We note that the metrics reported in [30], such as performance gap risk, which is EE model performance subtracted from full model performance, can easily be calculated using existing performance metrics, hence, we do not specifically report. Also, performance metrics are preferred for better and fair comparison.

**Setup:** For the training phase, we augment the pre-trained models with a linear output layer to serve as an exit. For the architecture of the $g$, we use a single linear layer with shared parameters across the exits that serves as a risk predictor. Following the existing methods [8], the inference is performed on a per-instance basis, setting the batch size to 1. The value of $\lambda$ is fixed to $\frac{\epsilon}{L}$, while the set of candidate thresholds is chosen as ten equally spaced values between 0.5 and 1.0 (including 1.0). An ablation study over different $\lambda$ values is also performed in Appendix B.3. It aligns with scenarios where low-latency is critical, such as processing individual requests from different users [42]. Similar to other baselines [30], we also choose $\epsilon = 0.01$ for all the experiments. An ablation study over $\epsilon$

| Models | COCO Karpathy test | | | | VQAv2 | VisDial | Spd. |
|---|---|---|---|---|---|---|---|
| | B4 | C | S | M | Acc | MRR | |
| BLIP-2-V-F | 44.0 | 145.8 | 25.3 | 31.9 | 81.8 | 45.9 | 1.00× |
| *EE-based models* | | | | | | | |
| PABEE-BLIP | 35.8 | 122.6 | 21.9 | 26.5 | 75.2 | 36.7 | 1.55× |
| ZTW | 36.3 | 121.1 | 21.7 | 26.0 | 76.9 | 37.8 | 1.59× |
| MuE | 37.5 | 126.8 | 22.4 | 27.7 | 78.0 | 38.4 | 1.49× |
| JEI-DNN | 38.2 | 128.4 | 22.6 | 28.0 | 78.4 | 38.1 | 1.63× |
| BEEM | 40.7 | 135.9 | 23.8 | 28.9 | 79.6 | 39.6 | 1.68× |
| *Risk-based models* | | | | | | | |
| CALM | 41.8 | 139.3 | 24.4 | 29.1 | 79.8 | 41.9 | 1.46× |
| MIP LAP | 41.2 | 137.4 | 24.1 | 28.9 | 78.6 | 40.2 | 1.55× |
| EERS | 42.5 | 140.3 | 24.7 | 29.5 | 80.0 | 42.6 | 1.38× |
| **Ours** | **43.3** | **142.9** | **25.0** | **31.1** | **81.1** | **42.9** | **1.72×** |

Table 3: Results on vision-language tasks (Visual Question Answering (VQA), test split of COCO and VisDial for visual dialogue) on BLIP-2-ViT-FlanT5-xl model with average speedup (Speedup).

| Models | in-domain | | near-domain | | out-domain | | full-dataset | | Speedup |
|---|---|---|---|---|---|---|---|---|---|
| | C | S | C | S | C | S | C | S | |
| BLIP-2 ViT FT5 | 123.7 | 16.3 | 120.2 | 15.9 | 124.8 | 15.1 | 121.6 | 15.8 | 1.00× |
| *EE-based models* | | | | | | | | | |
| PABEE-BLIP | 117.6 | 15.2 | 114.2 | 14.6 | 117.4 | 14.2 | 113.2 | 14.5 | 1.48× |
| ZTW | 119.1 | 15.5 | 115.4 | 14.8 | 119 | 14.4 | 115.9 | 14.8 | 1.35× |
| MuE | 118.3 | 15.3 | 114.8 | 14.7 | 119.5 | 14.4 | 115.6 | 14.7 | 1.62× |
| JEI-DNN | 119.8 | 15.5 | 116.2 | 15.0 | 119.9 | 14.6 | 116.1 | 14.9 | 1.58× |
| BEEM | 120.2 | 15.7 | 117.5 | 15.3 | 119.6 | 14.5 | 116.8 | 15.1 | 1.51× |
| *Risk-based EE methods* | | | | | | | | | |
| CALM | 121.0 | 15.9 | 117.8 | 15.5 | 119.8 | 14.7 | 117.1 | 15.2 | 1.47× |
| MIP LAP | 120.6 | 15.8 | 117.2 | 15.2 | 119.3 | 14.3 | 116.9 | 15.1 | 1.53× |
| EERS | 121.4 | 16.0 | 117.6 | 15.4 | 120.2 | 14.7 | 117.3 | 15.3 | 1.38× |
| **Ours** | **122.5** | **16.2** | **118.9** | **15.7** | **122.7** | **15.0** | **119.5** | **15.6** | **1.75×** |

Table 4: Results of BLIP-2-ViT-FlanT5-xl model on Nocaps dataset.

can be found in Appendix B.5. More hyperparameter details can be found in Table 11 and 12 in the Appendix.

**Baselines:** We consider three types of baselines to compare our model: *1) Full model:* This is the baseline showing conventional DNN performance. *2) Early Exit models:* These are the baselines where we consider various early exit methods, such as PABEE [52], patience-based exiting, and ZTW [49], on the other hand, uses aggregation of confidence scores across the exits. MuE [46] uses the similarity score of the hidden representations to decide exiting. JEI-DNN [18] learns a gating function to performs an exit. Finally, BEEM [8] is a method that utilizes ensemble methods to decide exiting. *3) Risk-based methods for EEs:* In this, we consider methods that consider the risk factor in the EE models. CALM [41] considers the risk of the T5-large model with EEs for text generation tasks, we extend this to other tasks as well. MIE LAP [36] considers aggregating the scores across exits to minimize risk and fix the overconfidence issue in the image classification tasks; we extend this to other tasks as well. Finally, we have EERC [30] that theoretically provides a simple extension of EE models to provide a method to choose the value of the exit threshold such that the risk is minimized. We use the same code and hyperparameters without any changes to get the results of the baselines.

**Results on Text classification:** In Table 1, we provide results on the text classification tasks (GLUE datasets) and the Question answering task over the BERT-large model. We observe that our method performs better than all the existing methods, where sometimes the performance of our method is better than the full model performance due to the impact of overthinking [32, 52]. The EE-based methods have more focus on the speedup part while having a higher loss in accuracy. An important observation is that BEEM, the EE-based method, outperforms risk-based methods due to its ensemble methods for performing an inference. However, this is only observed in easier tasks such as text classification. On the other hand, the risk-based methods have a lower loss in accuracy but observe a hit in speedup as their focus is solely on reducing the risk, while achieving minimal efficiency gains.

| Model | Noise = 0.1 | | | Noise = 0.5 | | | Noise = 1.0 | | |
|---|---|---|---|---|---|---|---|---|---|
| | B4 | C | Speed | B4 | C | Speed | B4 | C | Speed |
| Full model | 43.1 | 142.5 | 1.00× | 39.8 | 133.2 | 1.00× | 36.4 | 128.6 | 1.00× |
| EE-based methods | | | | | | | | | |
| JEI-DNN | 36.7 | 128.9 | 1.58× | 31.2 | 107.5 | 1.68× | 26.1 | 93.9 | 1.74× |
| BEEM | 38.6 | 132.0 | 1.60× | 34.7 | 122.3 | 1.71× | 30.6 | 104.5 | **1.82×** |
| Risk-based methods | | | | | | | | | |
| CALM | 39.1 | 131.5 | 1.55× | 34.8 | 122.7 | 1.63× | 31.5 | 110.3 | 1.72× |
| EERS | 39.6 | 134.3 | 1.47× | 36.0 | 126.4 | 1.54× | 32.7 | 116.9 | 1.68× |
| **Ours** | **41.3** | **139.7** | **1.70×** | **38.2** | **130.5** | **1.77×** | **35.0** | **125.5** | 1.81× |

Table 5: Results on the COCO test split by adding different levels of distortions to the test images.

Our method balances risk and efficiency using the reward function and minimizes the risk while getting improved efficiency.

**Results on Language Modeling:** In Table 2, we report the results on tasks such as text summarization, Question-answering and Machine translation. Our method consistently outperforms existing baselines both in terms of performance and speedup, due to its dynamic adjustments of thresholds based on the test dataset. One key observation from the existing tasks is that a very high static threshold was also getting a 2 points ROUGE-L score drop with a higher loss in efficiency. Due to this fact, there is a performance drop in the risk-based as well as EE-based methods shown in Table 2. Risk-based methods have a lower drop in performance as compared to EE-based methods. Our method, on the other hand, dynamically adjusts the threshold and has an option to switch from a hard criterion to exit to a soft one, helping it balance accuracy and efficiency.

**Results on the vision-language tasks:** In Table 3 and 4, we report the results on various vision-language tasks using the BLIP-2 backbone with ViT-g as the encoder and FlanT5-xl as the decoder. We observe that the efficiency gains of our method are the highest with minimal performance drop as compared to other risk-based as well as EE-based methods. The results on the Nocaps dataset in Table 4 suggest that our method is robust to distribution changes as compared to other methods as its performance dip is minimal when the images are from near-domain or out-of-domain. This is an advantage of the online adaptation of the threshold, where it learns the threshold based on the incoming dataset distribution.

**Robustness to distribution changes:** In Table 5, we provide the BLEU-4 scores for image captioning tasks, where additional noise is added to the test split of the dataset. The model, BLIP-2-FlanT5-xl, was trained on pristine (undistorted) images, and during inference, we add different levels of distortions in terms of Gaussian noise with mean zero and standard deviation $\sigma$; a higher standard deviation will have more noise in the image. Noise in images is a common real-world scenario where the model suffers with a loss in performance as shown in the Table 5, when it is trained on pristine images. The results from Table 5 suggest that our method's performance loss is small compared to other methods. The risk-based methods, specifically the EERS method, show some robustness to the noise, but due to their assumption that the test dataset is representative of the validation dataset, their performance is also significantly low. The reason for our method performing better is its dynamic adaptation of the threshold based on the incoming dataset distribution instead of a fixed threshold as done by other methods.

## 5.1 Ablation Study

**Analysis of the overconfidence issue** In Figure 1, we provide the box plots for the true class confidence values across the layers of the BERT-large model on QNLI dataset; we only plot the initial 12 layers out of 24 layers. Observe from the figure that the initial 3 layers have not learnt much information, hence mostly distributed around 0.5. After that, we see that the box plots are raised to a higher value. However, observe that there are around 12% samples at the 4th layer that are highly confident (more than probability 0.7 assigned to the wrong class as it is a binary classification task) on the wrong class, i.e., the model assigns a higher weight in output probability to the wrong class. This necessitates checking the reliability of the confidence. If not properly monitored these many samples with exit the backbone due to this high fake confidence and cause overall performance drop to be significant.

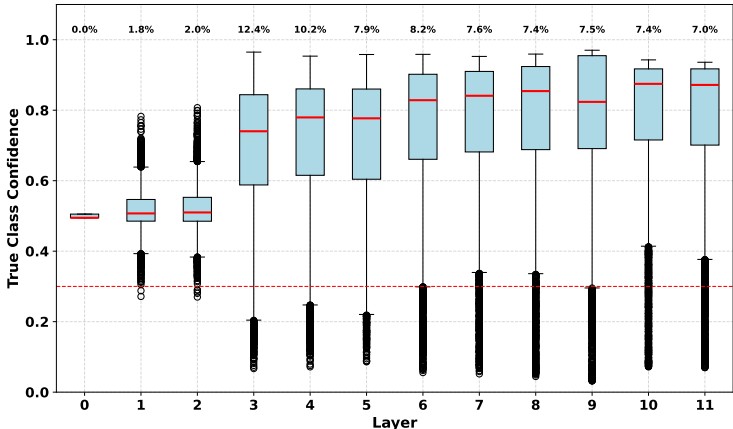

Figure 1: True class confidence on QNLI dataset across number of layers. The number on top of the plots show the percentage of samples below $0.3$ true class confidence. As it is a binary classification task, any sample below $0.3$ has a high confidence on the wrong class.

# 6 Conclusion

In this work, we introduced a risk-tolerant approach for Early-Exit Deep Neural Networks (EEDNNs) that balances accuracy and efficiency under dynamic test-time conditions. We first designed a reliability-driven confidence metric that quantifies the trustworthiness of exit predictions, and seamlessly integrated it into a reward-based formulation. To optimize this reward, we proposed an upper confidence bound based algorithm UAT, which dynamically selects the optimal exit threshold during inference, enabling the model to adapt to distributional shifts without requiring labels. We established a theoretical bound on the risk achieved by UAT. Experimental results on various tasks such as text classification, language modeling, and vision-language tasks further prove the effectiveness of our method. In this work, we considered a uniform exit threshold across all the exit layers. It is interesting to quantify the gains achievable by allowing the threshold to adapt to each layer.

# 7 Limitations

While our method effectively adapts exit thresholds based on prediction reliability, it relies on the quality of the learned confidence function $g$. Although all the existing methods use a global threshold across all the exits to assess the confidence, there is scope to have a local threshold for individual exits, then the model can be more efficient.

## Acknowledgements

Divya Jyoti Bajpai is supported by the Prime Minister's Research Fellowship (PMRF), Govt. of India. Manjesh K. Hanawal thanks funding support from SERB, Govt. of India, through the Core Research Grant (CRG/2022/008807) and MATRICS grant (MTR/2021/000645), and DST-Inria Targeted Programme. We also thank funding support from Amazon IIT-Bombay AI-ML Initiative (AIAIMLI).

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

# A  Appendix

## A.1  Proof of Lemma 3.1

Let $p(y|x)$ denote the probability of the true label $y$ given an input $x$. Consider an early-exit model that outputs a probability distribution $p(\cdot)$ over the set of classes for every threshold $\tau$. Let $p_\tau(y|x)$ be the true class probability given by the model, $p_\tau(\hat{y}|x)$ be the probability of prediction given by the model and $p_\tau(y = \hat{y})$ be the probability that predicted class is the true class.

For a fixed $\tau$, we can write the probability that the prediction made by the model is correct as:

$$p_\tau(y = \hat{y} \mid x) = p_\tau(y = \hat{y} \mid \hat{y}, x) \cdot p_\tau(\hat{y} \mid x)$$

$$= p_\tau(\hat{y}|x) \cdot p_\tau(y = \hat{y}|x, \hat{y})$$

$$\implies \arg\max_{\tau \in \Omega} p_\tau(y = \hat{y}|x) = \arg\max_{\tau \in \Omega} p_\tau(\hat{y}|x) \cdot p_\tau(y = \hat{y}|x, \hat{y})$$

Hence, we prove that the threshold $\tau$ that maximizes $p_\tau(\hat{y}|x) \cdot p_\tau(y = \hat{y}|x, \hat{y})$ value maximizes the chances of getting the prediction correct. $\square$

**Proof of Theorem 4.1:**

Before moving to prove the theorem, we will first bound the single-run regret of the UAT algorithm.

At each round $t = 1, 2, \ldots, T$, the algorithm selects an arm $\tau_t$ and receives a reward $r(\tau_t)$.

Regret Definition: We define the realized (actual) regret for a single run as:

$$R(T) = \sum_{t=1}^{T} (\mu^* - \mu_{A_t}) = \sum_{i=1}^{K} \Delta_i N_i(T),$$

where $N_i(T)$ is the number of times arm $i$ was pulled up to time $T$.

The UAT Algorithm: At each time $t$, the UAT algorithm selects the arm

$$\tau_t = \arg\max_i \left( \hat{\mu}_i(t) + \sqrt{\frac{2 \log T}{N_i(t)}} \right),$$

where: $\hat{\mu}_i(t)$ is the empirical mean reward of arm $i$ up to time $t$. $N_i(t)$ is the number of times arm $i$ has been pulled until time $t$.

High-Probability Bound Using Hoeffding's Inequality: By Hoeffding's inequality, for any $n \geq 1$ and any $\epsilon > 0$:

$$\mathbb{P}\left[|\hat{\mu}_i - \mu_i| \geq \epsilon\right] \leq 2 \exp(-2n\epsilon^2).$$

Define the confidence interval:

$$\text{CI}_i(t) = \left[ \hat{\mu}_i(t) - \sqrt{\frac{2 \log T}{N_i(t)}}, \hat{\mu}_i(t) + \sqrt{\frac{2 \log T}{N_i(t)}} \right].$$

In one particular round, true reward mean for $i$th arm lies in this interval with probability $1 - \delta$ where $\delta = 1/T^2$

Using the union bound over all arms and times, with probability at least $1 - 1/T$ that the true reward mean lies in the interval over all the rounds.

Bounding the Number of Suboptimal Pulls: Let us bound $N_i(T)$ for any suboptimal arm $i$ with $\Delta_i > 0$.

Suppose arm $i$ is selected at time $t$. For this to happen, the UAT index for arm $i$ must be at least as large as that for the optimal arm $i^*$:

$$\hat{\mu}_i(t) + \sqrt{\frac{2 \log T}{N_i(t)}} \geq \hat{\mu}_{i^*}(t) + \sqrt{\frac{2 \log T}{N_{i^*}(t)}}.$$

If all confidence intervals are valid (which occurs with high probability), then:

$$\mu_i + 2\sqrt{\frac{2 \log T}{N_i(t)}} \geq \mu^*.$$

Rearranging,

$$\sqrt{\frac{2\log T}{N_i(t)}} \geq \frac{\Delta_i}{2} \quad \Rightarrow \quad N_i(t) \leq \frac{8\log T}{\Delta_i^2}.$$

Thus, the number of times arm $i$ is pulled is at most:

$$N_i(T) \leq \left\lceil \frac{8\log T}{\Delta_i^2} \right\rceil + 1.$$

**Total Regret Bound:** Using $R(T) = \sum_{i:\Delta_i>0} \Delta_i N_i(T)$, we obtain:

$$R(T) \leq \sum_{i:\Delta_i>0} \Delta_i \left(\frac{8\log T}{\Delta_i^2} + 1\right) = \sum_{i:\Delta_i>0} \left(\frac{8\log T}{\Delta_i} + \Delta_i\right).$$

Finally: With probability at least $1 - \mathcal{O}(1/T)$, the realized regret of UAT satisfies:

$$R(T) \leq \sum_{i:\Delta_i>0} \left(\frac{8\log T}{\Delta_i} + \Delta_i\right) = \beta(T)$$

Let $p_\tau(\hat{y}|x)$ denote the model's confidence and $p_\tau(y = \hat{y}|x, \hat{y})$ its calibrated correctness probability. The empirical risk could be written as:

$$\hat{\mathcal{R}}(\pi) \triangleq 1 - \frac{1}{T}\sum_{t=1}^{T} p_\tau(y = \hat{y}|x, \hat{y})$$

The UAT algorithm that provides a threshold $\tau_t = \pi(x_t)$ using the policy $\pi$, for simplicity, we drop the index $t$. With probability atleast $\delta' = 1 - \frac{1}{T}$, the single run regret of UAT algorithm can be bounded as:

$$R(T) \leq \beta(T)$$

where the reward $r(\tau) = C_\tau^i \cdot (1 - C_g^i) - \lambda \cdot i = p_\tau(\hat{y}|x) \cdot p_\tau(y = \hat{y}|x, \hat{y}) - \lambda i$ as we consider $(1 - C_g^i)$ approximates the $p_\tau(y = \hat{y}|x, \hat{y})$ with probability $\delta_1$. Choose $\delta = (1 - \delta_1)(1 - \delta')$ then with probability $\delta$:

$$\sum_{t=1}^{T} [p_{\tau^*}(\hat{y}|x)p_{\tau^*}(y = \hat{y}|x, \hat{y}) - p_\tau(\hat{y}|x)p_\tau(y = \hat{y}|x, \hat{y}) - \lambda(i - i^*)] \leq \beta(T) \tag{5}$$

Rearranging terms and bounding exit differences $|i - i^*| \leq L$:

$$\sum_{t=1}^{T} (p_{\tau^*}(\cdot) - p_\tau(\cdot)) \leq \beta(T) + \lambda L T \tag{6}$$

where $p_\tau(\cdot) \triangleq p_\tau(\hat{y}|x)p_\tau(y = \hat{y}|x, \hat{y})$.

Using $p_\tau(\hat{y}|x) \leq 1$:

$$\sum_{t=1}^{T} p_\tau(y = \hat{y}|x, \hat{y}) \geq \sum_{t=1}^{T} p_\tau(\cdot) \geq \sum_{t=1}^{T} p_{\tau^*}(\cdot) - \beta(T) - \lambda L T$$

Dividing by $T$ and substituting into the risk definition:

$$\hat{\mathcal{R}}(\pi) \leq 1 - \frac{1}{T}\sum_{t=1}^{T} p_{\tau^*}(\cdot) + \frac{\beta(T)}{T} + \lambda L$$

As we assume that $\mathcal{R}^* = 1 - \frac{1}{T}\sum_{t=1}^{T} p_{\tau^*}(\cdot) \leq \epsilon_0$

$$\hat{\mathcal{R}}(\pi) \leq \epsilon^* + \frac{\beta(T)}{T} + \lambda L$$

For large $T$, $\frac{\beta(T)}{T} \to 0$ (since $\beta(T) = \mathcal{O}(\log T)$). To ensure $\hat{\mathcal{R}}(\pi) \leq \epsilon$:

$$\lambda L \leq \epsilon^d - \epsilon_0 \implies \lambda \leq \frac{\epsilon^d - \epsilon^*}{L} = \frac{\epsilon}{L}$$

Therefore, if we select $\lambda \leq \frac{\epsilon}{L}$, then with probability at least $\delta = (1 - \delta)(1 - \delta')$, the empirical risk $\hat{\mathcal{R}}(\pi)$ is bounded by $\epsilon$. This completes the proof. $\qquad \square$

| Model | SamSum | | COCO | | SQuAD | |
|---|---|---|---|---|---|---|
| | ROUGE-L | Speedup | B4 | Speedup | F1 | Speedup |
| $C_\tau^i$ | 46.49 | 1.43× | 40.72 | 1.62× | 81.9 | 1.57× |
| $C_\tau^i - \lambda \cdot i$ | 44.81 | 1.70× | 39.24 | 1.79× | 80.6 | 1.62× |
| $1 - C_g^i$ | 48.16 | 1.29× | 43.38 | 1.35× | 82.3 | 1.48× |
| $(1 - C_g^i) - \lambda \cdot i$ | 47.92 | 1.68× | 42.27 | 1.56× | 81.7 | 1.68× |
| $C_\tau^i(1 - C_g^i)$ | 48.34 | 1.35× | 43.51 | 1.48× | 83.3 | 1.60× |
| $C_\tau^i(1 - C_g^i) - \lambda \cdot i$ | 48.09 | 1.98× | 43.32 | 1.79× | 83.2 | 1.82× |

Table 6: Performance using different components in the reward function.

| $\epsilon$-values | $\epsilon$=0.01 | | $\epsilon$=0.05 | | $\epsilon$=0.1 | |
|---|---|---|---|---|---|---|
| Model | Risk | Speed | Risk | Speed | Risk | Speed |
| CALM | 04.3 | 1.37× | 04.9 | 1.54× | 05.3 | 1.92× |
| MIPLAP | 08.1 | 1.76× | 08.6 | 1.87× | 09.7 | 2.04× |
| EERS | 04.2 | 1.68× | 04.4 | 1.79× | 05.8 | 2.13× |
| **Ours** | **01.5** | **2.08×** | **02.3** | **2.25×** | **03.1** | **2.52×** |

Table 7: The impact on performance gap risk (in %) and speedup when the values of $\epsilon$ are varied.

# B  More Ablation study

## B.1  Importance of components of reward function

In Table 6, we show the performance of our model on different tasks and datasets when different components of the reward function given in equation 4 are used. The tasks used are summarization on the SamSum dataset using T5-large, image captioning on the COCO test split using BLIP-2-FlanT5-xl and Question Answering task using SQuAD on the BERT-large model. We observe that simply maximizing the prediction confidence term $C_\tau^i$ might lower the performance due to riskier exits while maximizing only the correctness probability $(1 - C_g^i)$ has better improvements, but causes underconfident exits as in such cases the model might be less confident, increasing the chances of wrong predictions even with good correctness probability. However, when we maximize the product, it becomes more robust as model confidence and reliability on confidence are both maximized, giving us better performance. As we add the penalty term, there is a slight drop in performance but a boost in efficiency, due to which we select it as the final objective in our setup.

## B.2  The value of $c$

The choice of coverage threshold $c$ depends on the prior knowledge of how complicated the dataset is based on the given distribution and model, as its task is to make $g$ produce higher values for a fraction of samples that are easy for the model. We do not a priori know this information, hence we rely on a validation dataset to assess the task complexity. Also, as the exits might find the sample easy or hard based on its information that might be different for different exits. Hence we choose a $c$ dependent on $i$, denoted as $c_i$. During training of the first epoch, we do not have any a prior knowledge of the task complexity hence we set $c_i$ to be 0 for all $i$, but after the first epoch we define $c_i = \frac{1}{N} \sum_{j=1}^N \mathbb{1}_{\{\hat{y}_{ij}=y\}}$ where $\hat{y}_{ij}$, i.e., the fraction of samples for which the exit is correct. This assesses the task complexity, making the $g$ function produce higher scores based on complexity.

## B.3  Risk-Efficiency trade-off

In Figure 2b, we show the risk-efficiency trade-off by plotting the percentage of performance drop as compared to the final layer vs Speedup for the SQuAD dataset on the BERT-large model by changing the parameter $\lambda$. For other baselines, we change their trade-off parameter and plot the results. The results suggest that our method consistently has lower risk while having a higher speedup. While we keep the value of $\epsilon = 0.01$, i.e, the percentage of $\mathcal{R}^G \leq \epsilon$. Our method keeps the risk lower than 1% till a speedup of 2.3× after which the risk crosses the desired bound, while other risk-based methods are very less tolerant to keep risk below 1% when we increase in speedup even when we restrict their permissible risk limit to 1% similar to ours.

## B.4  Regret Analysis

In Figure 2a, we plot the average cumulative regret while choosing the thresholds during inference using different policies on the QNLI dataset. The results are average across 5 runs where in every run the dataset was randomly reshuffled and fed to the algorithm. We plot the cumulative regret of final layer when all the samples always exit from the final layer, randomly selected, where a random threshold was assigned to each sample and fixed

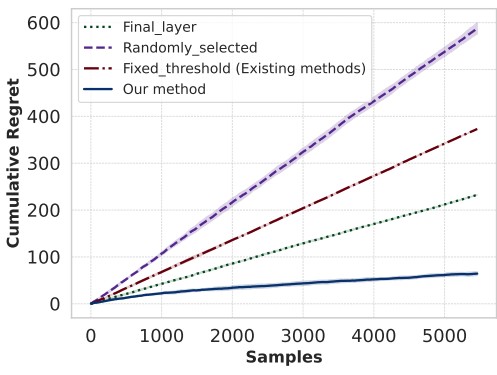
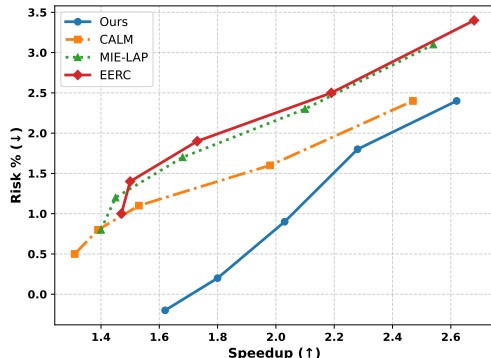

(a) Cumulative regret plots for different policies to choose the thresholds.

(b) Risk vs Speedup curve obtained by varying the trade-off parameter $\lambda$.

Figure 2: Comparative plots showing (a) cumulative regret for different threshold selection policies and (b) the risk-speedup trade-off.

threshold where a static threshold was utilized similar to existing EE and risk based methods. Our method observed the smallest regret as compared to others. This indicates the importance of learning a policy for the threshold selection instead of fixing it to a constant value.

### B.5  Changing $\epsilon$ values

In Table 7, we provide results on changing the $\epsilon$ that guides $\mathcal{R}^G \leq \epsilon$ on the SamSum dataset with T5-large model, note that, changing $\epsilon$ value changes the value of $\lambda$, allowing riskier samples to make an early exit. We report the percentage of increase in $\mathcal{R}^G$ as compared to the final layer. Observe from the table that even when more risky samples were allowed to exit early, our method's drop in performance is very low. This is due to an upgraded confidence metric that allows for an early exit only when it is confident and the confidence is also reliable. While imposing a hard criterion of $\epsilon = 0.01$, all the risk methods have violated that, while our being the closest to the required threshold. After relaxing the criteria, all methods except MIPLAP do not violate it, still, our method had the lowest performance gap risk as compared to other methods.

### B.6  Some important examples

In Table 10, we provide some examples on the classification datasets such as, SST-2 and QNLI, where we report the True label, Fake confidence (average model confidence over the true class), and risk score (average $C_g^i$). Observe from the Table that as the sample gets more ambiguous, the chances are high that the model outputs a wrong prediction. In such cases, it becomes important to flag those samples that is done by the $C_g^i$ value in our case. For instance, consider the sample "the under-7 crowd", which means that the review is negative, but the model probably understands the numbers as ratings and develops high confidence over the wrong class.

### B.7  Computational setup

In our experiments, we have used a setup of 5 NVIDIA A6000 GPUs. The highest run time was observed during training the BLIP-2 model for image captioning, which took 15 hours to train for 15 epochs. The average GPU runtime for the BLIP-2 model across tasks was 9 hours. The inference time for BLIP-2 was less than 20 minutes across all the tasks. For the T5-large model, the average runtime observed was 6 hours, with the highest on the Ques-

| Model | SamSum | QA | MT |
|---|---|---|---|
| T5-large | 1035s | 664s | 846s |
| T5-large w UAT | 1048s | 672s | 855s |

Table 8: Comparison of inference times (in seconds) across tasks on T5-large model.

tion Answering dataset due to a large number of epochs. For inference, it required less than 15 minutes across all datasets and tasks. Finally for the BERT-large model, the maximum runtime was on MNLI dataset with 1 hour of GPU runtime across epochs. The average runtime across GLUE tasks was 25 minutes. The average inference runtime was less than 2 minutes for all the tasks.

| Model | SamSum (T5) | COCO (BLIP-2) | QNLI (BERT) |
|---|---|---|---|
| **CE loss performance** | 48.84 | 43.9 | 90.6 |
| **Our loss performance** | 48.82 | 44.0 | 90.5 |

Table 9: Model performance trained using Conventional loss vs our loss.

| Example | True lbl. | Fake Conf. | $C_g^i$ |
|---|---|---|---|
| **SST-2** | | | |
| mysterious and brutal nature | positive | 0.79 | 0.68 |
| the under-7 crowd | negative | 0.91 | 0.95 |
| seems endless | negative | 0.90 | 0.82 |
| a movie with two stars | positive | 0.82 | 0.86 |
| **QNLI** | | | |
| Where was war fought? The war was fought primarily along the ... | entailment | 0.70 | 0.88 |
| What university donated the land ...? A site chosen in Houston, ... | entailment | 0.76 | 0.72 |

Table 10: Examples of samples achieving fake confidence, the table shows an example, its true label (True lbl.), the average confidence of the model on the wrong class (Fake Conf.) and the $(1 - C_g^i)$ score that abstains the sample early.

## B.8 Computational Complexity of UAT

Our method learns a policy to decide the exit threshold that is learned over time based on the data distribution. However, the computational complexity of UAT is negligible as it does not involve any big operations. In every, step, it requires maximizing over a small finite set (set of 10 elements) and maintain a list of reward functions. In Table 8, we perform experiment to show the additional complexity of our method. In this experiment exit was fixed to the final layer, i.e., there was no early exiting. We implement the UAT algorithm and do not change anything else, i.e., the exit is still from the final layer, but the only thing changed is that there is some additional computation happening due to UAT. So the time wall-clock time added to T5-large with UAT will be the computational complexity of UAT. Observe from Table 8, the computational complexity of UAT is negligible as the additional time is less than 15 seconds, further proving UAT's contribution during inference.

## B.9 Loss observations

In Table 9, we provide results of the final layer when the conventional cross-entropy loss is used to train the EEDNN i.e., just use the first term $\mathcal{L}_{CE}$ from Equation 3 and if you use the full loss function given in our method in Equation 3. The Table 9 suggests that adding the other components in the loss function does not impact the overall model performance. We check the final layer performance, as that would have been the most affected by the additional components. However, the effect was very insignificant, and we safely use the loss function given in Equation 3.

| Model | BLIP-2-FlanT5-xl | | BERT-large |
|---|---|---|---|
| **Tasks** | **Image captioning** | **VQA** | **GLUE** |
| **Finetuning epochs** | 15 | 5 | 5 |
| **Finetuning dataset** | Train Split | Train split | Train split |
| **Warmup steps** | 1000 | 1000 | 1000 |
| **Learning rate** | 1e-5 | 1e-5 | 1e-5 |
| **Batch size** | 16 | 16 | 32 |
| **AdamW beta** | (0.9, 0.999) | (0.9, 0.999) | (0.9, 0.999) |
| **Weight decay** | 0.05 | 0.05 | 0.05 |
| **Drop path** | 0 | 0 | – |
| **Image resolution** | 360 | 490 | – |
| **Prompt** | "a photo of" | "Question:{} Answer" | – |
| **Inference beam size** | 5 | 5 | – |

Table 11: Hyperparameter details of the BLIP-2-FlanT5-xl and BERT-large model on various tasks.

| Model | T5-large | | |
|---|---|---|---|
| Tasks | Summarization | MT | QA |
| Finetuning epochs | 5 | 5 | 10 |
| Finetuning dataset | Train Split | Train Split | Train Split |
| Warmup steps | 1000 | 1000 | 1000 |
| Learning rate | 1e-4 | 1e-4 | 1e-4 |
| Batch size | 8 | 8 | 16 |
| AdamW beta | (0.9, 0.999) | (0.9, 0.999) | (0.9, 0.999) |
| Weight decay | 0.05 | 0.05 | 0.05 |
| Drop path | 0 | 0 | 0 |
| Image resolution | – | – | – |
| Prompt | summarize: | translate german to english | – |
| Inference beam size | 5 | 5 | 5 |

Table 12: Hypereparameters details on the T5-large model for various tasks.

