# OpenReview forum: "Beyond Greedy Exits: Improved Early Exit Decisions for Risk Control and Reliability"
_NeurIPS.cc/2025/Conference — NeurIPS 2025 poster_

### Official Review · Reviewer_NGvD · 2025-06-19

**Clarity:** 2
**Significance:** 1
**Originality:** 1
**Rating:** 2
**Confidence:** 5

**Summary:**

This paper proposes a reward function that assesses predictive certainty for early exit mechanism to balance computational efficiency and prediction quality. It utilizes a Multi-Armed Bandit framework to take into account it for an early exit decision. It evaluates the proposed method in terms of inference speed up and performance.

**Questions:**

The authors state “reliability of confidence” in multiple places, it is not clear what is the difference between “reliability of confidence” and “correctness/accuracy of confidence”. As far as I figure out, by reading the paper, it is ultimately the correctness of the confidence. Please clarify what the authors mean by “reliability.”

**Ethical Concerns:**

["NO or VERY MINOR ethics concerns only"]

**Final Justification:**

This work does not have a significant novelty.

**Limitations:**

* One of the limitations is the novelty – the problem is not novel.

* The another limitation is that the work contains conflicting assumptions and/or explanations with regard to the offline training mechanism (especially the pre-training mechanism) of the proposed Early exit mechanism. I do not believe it can be practically deployed in real-world applications.

**Paper Formatting Concerns:**

* Texts appear on the 10th page.

**Quality:**

2

**Strengths And Weaknesses:**

* The problem is not novel.

* “High confidence” depends on the threshold. If the model overall overaly confident, the threshold can be adjusted. Accordingly, depending on the threshold, the Fig. 2’s results can vary, and the authors’ claim on the over confidence can be not true.

* The statement in line 59, “During inference, the data arrives in an online fashion.” is not always true. It depends on the applications/systems/ downstream tasks.

* Thue authors state “reliability of confidence” in multiple places, it is not clear what is the difference between “reliability of confidence” and “correctness/accuracy of confidence”. As far as I figure out, by reading the paper, it is ultimately the correctness of the confidence. Please clarify what the authors mean by “reliability.” – Which situation would be reliable but incorrect and which situation would be not reliable but correct?

* There is a conflicting or confusing point. The authors mentioned “During inference, the data arrives in an online fashion” in line 59. But, in line 160, for the first stage, they mention “we perform offline training of the EEDNN…”. If the model is already deployed to receive data for inference arriving an online fashion, how it can be offline-trained again? Then, the model cannot accommodate such online/realtime data, and also that is against the aim of the EEDNN for inference.

---

> ### Author Response · Authors · 2025-07-31
> **Rebuttal**
>
> Thanks for your time and effort in reviewing our work, we address the concerns below:
>
> **Q1:** The problem is not novel.
>
> **A1:** We respectfully disagree with the assertion that our problem lacks novelty. Though reliability in model outputs has been explored broadly [1, 2], it has been applied to Early Exit (EE) models only recently [3, 4] (in 2024). EE frameworks bring the additional challenge of handling trade-offs between speed and accuracy. [3, 4] work under the assumption that test-time distributions match those seen during training—a condition often violated in real-world deployments. Our work bridges this gap.
> Our method directly addresses this limitation by introducing an adaptive thresholding framework that dynamically adjusts to input distributions while simultaneously evaluating the reliability of the model’s confidence outputs. This enables the system to maintain a favourable accuracy-latency trade-off without relying on static, offline-tuned thresholds, which are often brittle and suboptimal in real-world deployments.
>
> To the best of our knowledge, no prior work has considered adaptive thresholding for early exits with explicit modeling of confidence reliability. We have highlighted these issues in the paper, and developed a framework and algorithm that outperforms all existing static solutions.
>
> That said, we would appreciate it if the reviewer could pinpoint what makes our solution not novel, like pointing to 1) Prior works that address the problems we tackle in EE models or 2) any existing adaptive thresholding methods that also account for confidence reliability under distribution shifts .
>
> This will help us better understand your novelty perspective and better position our paper.
>
>
>
>
> **Q2:** “High confidence” depends on the threshold. If the model overall overaly confident, the threshold can be adjusted. Accordingly, depending on the threshold, the Fig. 2’s results can vary, and the authors’ claim on the over confidence can be not true.
>
> **A2:** We agree that one can increase the confidence threshold to counteract model overconfidence. But this is not a good idea as it undermines the core purpose of early exiting to optimally trade off accuracy and computation/speed. Instead, our method handles it by introducing two new aspects 1) reliability metrics and 2) dynamic thresholding. For reliability, we identify samples where the model is overconfident on incorrect predictions and downscale their confidence scores 2) In dynamic thresholding, we learn an appropriate threshold that is optimal for the observed data during inference. This targeted approach preserves efficiency while improving reliability.
>
> We believe Figure 2 is misinterpreted.  It depicts the distribution of confidence scores across exits without applying any threshold. Hence these plots are independent of the threshold values and Figure remains the same for any threshold. The reference line at 0.3 is not a threshold but a visual marker for the proportion of samples with confidence above 70% on incorrect classes. This visualization highlights overconfidence issues.
>
> **Q3**: The statement in line 59, “During inference, the data arrives in an online fashion.” is not always true. It depends on the applications/systems/ downstream tasks.
>
> **A3:** We believe it is line 49 not 59. Thanks for pointing it out. We see possibility of some confusion regarding the applicability of our method to batch inference due to use of word “online fashion.” Even in batch inference, each sample undergoes forward propagation individually, allowing the Multi-Armed Bandit (MAB) framework to update thresholds and policies on a per-sample basis. Therefore, our MAB-based approach naturally extends to batch settings, as it continuously adapts using feedback from each sample regardless of batch grouping. This makes our method effective and applicable in both strictly online and batch inference scenarios.
>
> We used the word ‘online’ as it is common in the MAB setups where the updates happen sequentially. But, as clarified above, our method works for both online and batch setup.
>
>
>
> [1] https://proceedings.neurips.cc/paper/2019/file/757f843a169cc678064d9530d12a1881-Paper.pdf
> [2] https://proceedings.mlr.press/v97/geifman19a/geifman19a.pdf
> [3] https://proceedings.neurips.cc/paper_files/paper/2024/file/ea5a63f7ddb82e58623693fd1f4933f7-Paper-Conference.pdf
> [4] https://openaccess.thecvf.com/content/WACV2024/papers/Meronen_Fixing_Overconfidence_in_Dynamic_Neural_Networks_WACV_2024_paper.pdf
>
>
>
> Further clarification can be found below.

---

> ### Author Response · Authors · 2025-07-31
> **Rebuttal**
>
> **Q4:** Thue authors state “reliability of confidence” in multiple places, it is not clear what is the difference between “reliability of confidence” and “correctness/accuracy of confidence”. As far as I figure out, by reading the paper, it is ultimately the correctness of the confidence. Please clarify what the authors mean by “reliability.” – Which situation would be reliable but incorrect and which situation would be not reliable but correct?
>
> **A4**: Yes, you are correct. By “reliability of confidence,” we essentially mean how much we can trust or depend on the confidence score provided by the model. In other words, the correctness of that confidence.
>
> **To clarify, consider this simple example:** if a model predicts a class with 90% confidence, but the prediction is wrong, then the confidence is not reliable (in this case, the sample will be flagged by the g function), it is overconfident despite being incorrect. Conversely, if a model outputs 90% confidence and the prediction is indeed correct, then the confidence is reliable.
> So, “correctness of confidence” and “reliability of confidence” are actually the same, and hence we have used them interchangeably. There is nothing that is reliable but incorrect and not reliable and correct in our setup. Our work focuses on quantifying and improving this reliability.
>
>
> **Q5:** There is a conflicting or confusing point. The authors mentioned “During inference, the data arrives in an online fashion” in line 59. But, in line 160, for the first stage, they mention “we perform offline training of the EEDNN…”. If the model is already deployed to receive data for inference arriving an online fashion, how it can be offline-trained again? Then, the model cannot accommodate such online/realtime data, and also that is against the aim of the EEDNN for inference.
>
> **A5:** Again, we believe you meant Line 49 not 59! Note that Line 49 is in the introduction where we are giving an overall view of our work. Here we are talking about the inference stage. And, yes, we are learning in the inference stage using the MAB setup to adapt to the incoming samples!
>
> Whereas Line 160 is talking about how the model is prepared for the deployment. Here classical offline training is applied. It is important to note that, as highlighted at multiple places, our methods bring in this approach of dynamically adapting thresholds during inference, which tentamounts to learning the right confidence threshold during the inference process. In nutshell, training happens pre-deployment and post deployment. But what is trained is different. During predeployment, EE classifiers are tuned with off-line data and during post deployment, the confidence thresholds are learned using the observed unlabelled data.
> To further clarify, our method consists of two distinct phases:
>
> 1) Offline Training Phase (Pre-deployment): Here, the Early Exit Deep Neural Network (EEDNN) model is fully trained using traditional offline supervised learning on training data. This training sets the model weights and prepares it for deployment. Note that all EE-based methods undergo training.
>
>
> 2) Dynamic Adaptation Phase (Post-deployment): After deployment, the model weights remain fixed; no further model training or weight updates occur. Instead, our method dynamically adapts the exit thresholds using a Multi-Armed Bandit (MAB) framework. The MAB algorithm updates the thresholds to optimize performance, enabling the model to handle distribution shifts and maintain reliability in real time.
>
>
> Thus, the offline training refers to preparing the model before deployment, while the “online fashion” refers specifically to threshold adaptation during inference.
>
> **In crux dynamic adaptation and reliability aspects are important issues that our work addresses that were not considered in earlier works. We hope this clarification changes your view of the motivation and contribution of our work.**
>
> We again thank you for your time and efforts in reviewing our work. We hope that we clarified most of your doubts, if you have any further question, please let us know else, **we sincerely request you to reassess the scores**.

---

> ### Comment · Reviewer_NGvD · 2025-08-04
>
> I appreciate the response by the authors. Unfortunately, the novelty issue is fundamental and serious. The problem this paper tries to solve has been in the community for a while, almost a decade, and many papers have addressed the problem. This is why the submitted paper **lacks the novelty**. Here are some examples of them. Please refer to these earlier works:
>
> * Adaptive Neural Networks for Efficient Inference, ICML 2017
> * Spatially Adaptive Computation Time for Residual Networks, CVPR 2017
> * SkipNet: Learning Dynamic Routing in Convolutional Networks, ECCV 2018
> * Shallow-Deep Networks: Understanding and Mitigating Network Overthinking, ICML 2019
> * EPNet: Learning to Exit with Flexible Multi-Branch Network, CIKM 2020
> * AdaEE: Adaptive Early Exit DNN Inference Through Multi Armed Bandits, IEEE ICC 2023
> * Adaptive Deep Neural Network Inference Optimization with EENet, WACV, 2024
>
> [Survey]
> * Adaptive Inference through Early Exit Networks (survey) – Laskaridis et al., Workshop at EMDL 2021  - This is a survey paper, but not an existing paper overlapping your work. I am posting this for your information.
>
> Because of the **novelty issue**, I maintain my original rating.

---

> ### Author Response · Authors · 2025-08-04
> **Further clarifications**
>
> We sincerely thank the reviewer for pointing out important prior works. We would like to clarify our position regarding the novelty and contributions of our paper.
>
> First, we acknowledge that the problem of adaptive inference via early exits has been studied extensively in the literature, and we appreciate the list of references provided. In fact, most of these works are already cited in our paper (e.g., shallow-deep) and we are well-aware of them. We explicitly position our work within this literature and emphasize how our approach differs in both methodology and scope.
>
> **Please note that none of these works (given by the reviewer) consider the risk aspects as done in [1] [2] and ours.**
>
> However, we respectfully argue that the existence of prior work on a general problem does not preclude the novelty of new contributions that advance the field—particularly when:
>
> 1) we introduce new theoretical insights,
>
> 2) propose improved algorithms with better practical trade-offs, and
>
> 3) demonstrate superior empirical performance in relevant scenarios.
>
> Our work clearly separates itself from prior approaches by proposing an improved decision criterion for early exits considerign not only confidence but also the reliability of confidence. To the best of our knowledge, none of the cited prior work (given by thr reviewer) explicitly accounts for the reliability of the confidence outputs from the model—a well-known and important issue in deep learning. In fact, this has been identified as a critical problem in Early Exits [1], [2] (2024). We build upon and enhance these approaches, leading to a significant reduction in the risk of incorrect predictions while simultaneously achieving better speedups as compared to them.
>
> **Naturally, any work aimed at improving early-exits will share some thematic overlap with existing early-exit architectures. However, the core novelty of our method lies in revisiting and improving the risk in Early Exit models, which, to the best of our knowledge, has not been effectively addressed in this context. We again request the reviewer to have a look at the cited papers below as these consider a similar concern we are raising and solving in a much efficient and better way**
>
> [1] https://proceedings.neurips.cc/paper_files/paper/2024/file/ea5a63f7ddb82e58623693fd1f4933f7-Paper-Conference.pdf
>
> [2] https://openaccess.thecvf.com/content/WACV2024/papers/Meronen_Fixing_Overconfidence_in_Dynamic_Neural_Networks_WACV_2024_paper.pdf

---

> > ### Author Response · Authors · 2025-08-07
> > **Gentle reminder**
> >
> > Dear reviewer NGvD,
> >
> > We hope that you reviewed the additional clarifications we provided in response to your follow-up comments. We sincerely hope that our explanations have addressed the majority of your concerns. If there are any remaining questions or points that require further clarification, we would be more than happy to elaborate.
> >
> > Otherwise, we sincerely request that you consider reassessing your evaluation in light of the clarifications provided. Once again thank you for your time and effort in reviewing our work.

---

> > > ### Comment · Reviewer_NGvD · 2025-08-09
> > >
> > > Thank you for your response. Unfortunately, I am not able to agree with the authors. The authors stated that the submitted work is to propose “improved algorithms” and “revisit and improve the risk in Early Exit models”. Then, that means the problem itself is not novel enough. I even see an ambiguous distinction between the submitted work and [1] & [2]. In particular, the submitted work is dominantly following the methods of [2]. Even [2] already provides the theoretical foundation. Also, the submitted work just took the scoring mechanism of [1] and extended it to other tasks.
> > > Besides that, I would like to point out an overclaiming issue. I believe the authors are overclaiming by stating “”most” of these works are already cited in our paper”. As I can see, only one paper out of 8 papers is cited. It is not “most”. This is another issue with the submitted work since the work does not have a good grip on the closely related literature.
> > >
> > > As a result, in my honest opinion, unfortunately, this is not an inspiring work in terms of novelty, so I cannot raise my score.
> > >
> > > [1] Fixing Overconfidence in Dynamic Neural Networks, WACV 2024
> > >
> > > [2] Fast yet Safe: Early-Exiting with Risk Control, NeurIPS 2024

---

> ### Author Response · Authors · 2025-08-09
> **Further clarification on comments of Reviewer NGvD**
>
> **Q1: improved algorithms” and “revisit” means the problem itself is not novel enough.**
>
> **A1:** We strongly disagree that 'revisiting' and 'improving' exiting work precludes the novelty of any work. In fact, several of the works in  NeurIPS improve and revisit the problems. Below are a few papers that make it explicit in the title itself
>
> [1] "Improved Algorithms for Linear Stochastic Bandits" NIPS 2011
>
> [2] "Revisiting Neural Scaling Laws in Language and Vision" NeurIPS 2022
>
> [3] "Revisit the Power of Vanilla Knowledge Distillation: from Small Scale to Large Scale", NeurIPS 2023
>
> [4] "Improved Regret for Bandit Convex Optimization with Delayed Feedback" NeurIPS 2024
>
> Often, existing works are revisited to address gaps with new ideas that bring a fresh perspective to the problem. It is the case in our paper, where we address the reliability issues that could arise due to distribution shifts in practical deployment scenarios.
>
> [1] https://papers.nips.cc/paper_files/paper/2011/file/e1d5be1c7f2f456670de3d53c7b54f4a-Paper.pdf
>
> [2] https://proceedings.neurips.cc/paper_files/paper/2022/file/8c22e5e918198702765ecff4b20d0a90-Paper-Conference.pdf
>
> [3] https://proceedings.neurips.cc/paper_files/paper/2023/file/204f828ba287fdecf41dd002e9a07d8c-Paper-Conference.pdf
>
> [4] https://neurips.cc/virtual/2024/poster/94556
>
> **Q2: ambiguous distinction between the submitted work and [1] & [2]**
>
> **A2:** We have highlighted the distinction already in the related works section. For example, in line 97-105, we discussed all the distinctions. Note note that authors in [2] remark that their method does not deal with distribution shifts and highlight it as the limitation of their work, leaving it as a future work (refer to limitation and future work section in [2])  . Below, we again highlight distinctions
>
> **Fixed threshold:** Both [1] and [2] assume that the test data distribution is the same as the training data distribution to learn a fixed threshold. We remove this assumption in our work.
>
> **Theoretical bounds:** The guarantees in [2] hold under the explicit assumption that validation data has the same distribution as the test data. Our guarantee does not need this assumption. [1] does not provide any guarantees.
>
> **Scoring function:**  By the scoring function, we assume that the reviewer is referring to the function used to decide exiting. We believe it is misunderstood, as the reviewer ignored the reliability metric in the scoring function. Below we reproduce the scoring functions for comparison.
>
> The scoring function in [1] :
>
> $S_k:=p_k^{ens}(\hat{y}_i | x_i)$
>
> $=\frac{1}{\sum_{l=1}^{k} w_l } (\sum_{m=1}^{k} w_m p_m(\hat{y}_i | x_i) $
>
> where ${p}_k(\hat{y}_i | x_i))$ is the probability estimate on the predicted class i.e., max of the probability distribution output from the $k$th layer over the classes and $p_k^{ens}$ is the weighted average.
>
> The scoring function in our work is
>
> $S_k = C^k \cdot (1-C_g^k)$
> where $C^k$ is the max probability of the layer $k$ (same as $p_k$)  and $(1-C_g^k)$ is the function that considers the reliability of $C^k$.
>
> Note that the scoring function in [1] does not have have reliability component in it. It only aggregates the confidence scores (probability estimates) across the layers. **Whereas, we do not aggregate, but discount the probability estimates appropriately with a reliability score. Discounting the reliability score is the new idea introduced in our scoring function to handle the distributional shifts.**
>
>
>
> **Q3: work does not have a good grip on the closely related literature.**
>
> **A3:** We disagree; we have followed the literature closely.  We cited a recent survey paper that gives details of work on EE models ([11] in paper), that includes most of the papers highlighted by the reviewers like EPNET, UCBEE (extension of AdaEE). We did not cite all of them again, but focused more on the recent works. We did miss the recent work on EENet (WACV, 2024), but it did not consider any risk factor which is the focus of our work.
>
>
> In addition to [1] and [2] we compared our work with other most recent papers as listed below:
>
> PABEE: https://proceedings.neurips.cc/paper_files/paper/2020/file/d4dd111a4fd973394238aca5c05bebe3-Paper.pdf. (Neurips 2020)
>
> Zero-Time Waste: https://papers.neurips.cc/paper_files/paper/2021/file/149ef6419512be56a93169cd5e6fa8fd-Paper.pdf. (Neurips 2021)
>
> MuE: https://openaccess.thecvf.com/content/CVPR2023/papers/Tang_You_Need_Multiple_Exiting_Dynamic_Early_Exiting_for_Accelerating_Unified_CVPR_2023_paper.pdf. (CVPR 2023)
>
> JEI-DNN: https://arxiv.org/pdf/2310.09163. (ICLR 2024)
>
> BEEM: https://arxiv.org/pdf/2502.00745. (ICLR 2025)
>
> CALM: https://papers.neurips.cc/paper_files/paper/2022/file/6fac9e316a4ae75ea244ddcef1982c71-Paper-Conference.pdf. (Neurips 2022)
>
>
> Hence, we believe we have done our homework well and have a good understanding of the related literature.

---

### Official Review · Reviewer_Cxuf · 2025-07-02

**Clarity:** 4
**Significance:** 3
**Originality:** 3
**Rating:** 5
**Confidence:** 4

**Summary:**

The paper addresses the problem of selecting an appropriate exit threshold in early-exit neural networks. The core idea is to focus on confident but incorrect predictions. To tackle threshold selection, the authors introduce a formal notion of risk and apply a Multi-Armed Bandit approach guided by this notion. They support their method with theoretical justification and present empirical comparisons against state-of-the-art early-exit strategies.

**Questions:**

1. What is the definition of near-domain and out-of-domain in Section 5? Specifically, what techniques are used to generate samples that are not in the domain? Are there any limitations on how far the data can deviate from the original domain?
2. From my experience, CeeBERT [7] and UCBEE use Multi-Armed Bandit (MAB) techniques for exit decision-making. However, the authors do not compare their method with these or any other MAB-based approaches in the main results (e.g., Tables 1–5). If I’m mistaken, please correct me. Otherwise, it would be helpful to clarify whether there is a specific reason such a comparison was omitted. If not, including it would be strongly encouraged.
3. In my opinion, using a single threshold shared across all exits is a major limitation. Can the authors provide experiments with exit-wise thresholds? I understand that this may be challenging for networks with many intermediate classifiers. To make this more tractable, a simplified setup—such as a predefined threshold schedule (e.g., linear or exponential decay using a single additional parameter)—would already be valuable. An alternative approach would be to adopt dynamic thresholding, as proposed in Fixing Overconfidence in Dynamic Neural Networks [36], where a threshold vector is derived from a single “heaviness level” parameter. https://github.com/AaltoML/calibrated-dnn/blob/main/adaptive_inference.py#L251-L259

I would consider raising my score if the authors address the above concerns and provide additional clarification or experiments where suggested.

**Ethical Concerns:**

["NO or VERY MINOR ethics concerns only"]

**Final Justification:**

The authors addressed my concerns. This is a good piece of work.

**Limitations:**

Yes, the authors adequately addressed the limitations.

**Paper Formatting Concerns:**

Did not notice any.

**Quality:**

3

**Strengths And Weaknesses:**

Strengths:
1. The paper is well-structured, easy to follow, and well-motivated.
2. The definitions of the loss and reward functions are clear and intuitive.
3. The authors perform distribution shift experiments, which are crucial for practical applications.
4. The work includes multiple evaluations across various tasks.
5. The included mathematical theorems enhance the understanding of the topic.

Weaknesses:
1. There is no evaluation on vision-only tasks, such as CIFAR-100 or ImageNet-1k. Many works in the early-exit literature also consider these benchmarks.
2. Evaluation on CNNs would be beneficial as well, especially since all the models considered in the paper are transformers (which is understandable for NLP tasks)
3. The authors do not provide performance scores (e.g., accuracy) for individual heads – that is, what the accuracy would be if all samples exited at the i-th intermediate classifier – nor do they report the exit rates (i.e., the proportion of samples exiting at each head). Including such information would offer valuable insights into the model's decision-making process.
4. The authors consider a single-threshold strategy for exiting, which is known to be suboptimal. While it is commendable that the authors explicitly acknowledge this limitation, it may significantly affect the generalizability of the proposed improvements.

---

> ### Author Response · Authors · 2025-07-31
> **Rebuttal**
>
> Thanks for your postive remarks and insightful feedback.
>
> **Q1:** There is no evaluation on .....
>
> **A1:** Here are the results for the MobileNet-v2 model over the Cifar-100 dataset. We choose the MobileNet-v2 as it is a CNN based model. This answers both the question of how it can be applied to CNN models and the performance on image-only tasks.
>
> |              | Accuracy | Speed |
> | ------------ | -------- | ----- |
> | MobileNET-V2 | 75.3     | 1     |
> | BEEM         | 74.5     | 1.59  |
> | MIELAP       | 74.2     | 1.48  |
> | EERC         | 74.8     | 1.65  |
> | Ours         | 75.1     | 1.77  |
>
> The table above shows that our method performs well even in image-only tasks with a CNN architecture. We will also perform experiments on ImageNet-1k dataset and add them in the final version.
>
> The code for the above results can be found in the anonymous repo given below.
>
> https://anonymous.4open.science/r/Mobilenet_EE-3565/README.md
>
> **Q2:** The authors do not provide performance scores (e.g., accuracy) ....
>
> **A2:** Thank you for the suggestion. We agree that reporting per-head accuracy and exit rates offers a deeper understanding of the model's behaviour. Below, we report these statistics for the QNLI dataset using a BERT-large backbone. These results can be reproduced using our source code. We will include them in the final version of the paper.
>
> | Exit Layer       | 1   | 2   | 3   | 4   | 5   | 6   | 7   | 8   | 9   | 10  | 11  | 12  | 13  | 14  | 15  | 16  | 17  | 18  | 19  | 20  | 21  | 22  | 23  | 24  |
> |-----|-----|-----|-----|-----|-----|-----|-----|-----|-----|-----|-----|-----|-----|-----|-----|-----|-----|-----|-----|-----|-----|-----|-----|-----|
> | Accuracy (%)     | 49  | 51  | 59  | 65  | 71  | 74  | 76  | 79  | 81  | 83  | 84  | 85  | 85  | 86  | 87  | 88  | 88  | 89  | 91  | 93  | 93  | 92  | 92  | 92  |
> | Exit Rate (%)    | 0   | 0   | 0   | 2   | 5   | 7   | 8   | 9   | 6   | 11  | 5   | 6   | 4   | 5   | 3   | 2   | 1   | 2   | 4   | 3   | 2   | 1   | 1   | 13  |
>
> These values demonstrate how the model progressively improves in accuracy across layers. From exit rates, we observe that 1) Only 13% of the samples are processed till the final layer. 2) Another thing to note is: More than 50% samples do not traverse 50% of the backbone.
>
> **Q3:** The authors consider a single-threshold strategy for exiting, which is known to be suboptimal. While it is commendable that the authors explicitly acknowledge this limitation, it may...
>
> **A3:** We adopt a single-threshold strategy as it aligns with most existing risk-sensitive early-exit methods [1], [2]. Using a global threshold simplifies threshold selection and significantly reduces computational complexity. If we have to adapt different thresholds at different exits then the number of thresholds over which we have to learn can be large. For example, with 24 exits and 5 candidate thresholds per exit, a full grid search would require evaluating $24 \times 5 = 120$ combinations, which quickly becomes impractical as model sizes grow.
>
> Most prior works in Early Exit literature therefore, rely on either a fixed global threshold or heuristics like a decay parameter to adjust thresholds across exits. Our method supports such dynamic adjustments. As suggested, we produced the results for the exponential decay scheme. See details in Q6.
>
>
>
> Q4: What is the definition of near-domain and out-of-domain..
>
> A4: We use two datasets with distribution shifts: NoCaps dataset and synthetic dataset. NoCaps dataset is a standard dataset consisting of splits such as in-domain, near domain and out-of-domain. These terms are defined in [3] and we reproduce this below.
>
> **In-domain images** contain objects that directly overlap with the COCO dataset classes (training dataset), representing familiar content to the model.
>
> **Near-domain images** feature a combination of both in-domain objects and out-of-domain objects, representing a mix of familiar and novel elements.
>
> **Out-of-domain images** exclusively contain objects not present in COCO, challenging the model to generalize to unseen concepts.
>
> Models pretrained on datasets beyond COCO tend to perform reasonably well across all these domains.
>
> For synthetic dataset, we created dataset with distribution shift by adding Gaussian noise to the image data (described in lines 324–335), which is an additional experimental setup distinct from the NoCaps domain splits. The results on this are reported in table 5.
>
> Regarding how far the distributional shifts can happen, we have not thoroughly evaluated this, in our experiments we change the variance of gaussian distribution in the range 0.1 to 1.0.
>
> [1] https://proceedings.neurips.cc/paper_files/paper/2020/file/d4dd111a4fd973394238aca5c05bebe3-Paper.pdf
>
> [2] https://aclanthology.org/2025.findings-acl.1209.pdf
>
> [3] https://openaccess.thecvf.com/content_ICCV_2019/papers/Agrawal_nocaps_novel_object_captioning_at_scale_ICCV_2019_paper.pdf.
>
> Further clarifications in the next comment:

---

> ### Author Response · Authors · 2025-07-31
> **Further Rebuttal**
>
> **Q5:** From my experience, CeeBERT [7] and ..
>
> **A5:** Thank you for this observation. We did compare it in the ablation studies but do not explicitly mentioned the names.  In Table 6, the second row corresponds to the method used by CeeBERT. We note that our model captures CeeBERT by ignoring the reliability part i.e., CeeBERT’s reward function does not account for the reliability of confidence, unlike our full reward design, which makes this variant a natural part of our ablation study; hence, we consider it as an ablation test. Still, to avoid any ambiguity and improve clarity, we will explicitly label this entry as “CeeBERT” and apply this notation across all evaluated models in the final version.
>
> We did not compare to UCBEE as in UCBEE a sample can only exit through a fixed intermediate layer while in our setup, the sample can exit from any intermediate layer.
>
> **Q6:** In my opinion, using a single threshold shared across all exits is a major limitation. Can the authors provide experiments with exit-wise thresholds? I understand that this may be challenging for networks with many intermediate classifiers. To make this more tractable, a simplified setup—such as a predefined threshold schedule (e.g., linear or exponential decay using a single additional parameter)—would already be valuable. An alternative approach would be to adopt dynamic thresholding, as proposed in Fixing Overconfidence in Dynamic Neural Networks [36], where a threshold vector is derived from a single “heaviness level” parameter.
>
> **A6:** Thank you for your thoughtful suggestion. While exit-wise thresholds are an interesting direction, we believe using a single global threshold, as adopted by most existing early-exit methods, offers important practical advantages in both simplicity and generalization.
>
> We agree that parameterized threshold schedules (e.g., exponential decay) offer a good compromise between complexity and performance. Our framework is flexible enough to support such strategies, and we have incorporated your suggestion in our experiments. Specifically, after the UCB algorithm selects a base threshold $\lambda_{\text{base}}$, we apply an exponential decay as follows:
>
> $$\lambda_i = \max \left( \lambda_{\text{base}} \cdot \text{decay}^i, \quad \lambda_{\text{base}} - 0.05 \right)$$
> Here, $i$ denotes the exit layer index. The threshold for deeper exits decreases exponentially but is clipped to remain within 0.05 of the base threshold. This clipping prevents thresholds from becoming too low, which could lead to premature or immature exits. The results for T5 model and BERT are given below:
>
> T5-model:
>
> |                | Summarization | QA    | MT    | Speedup |
> | -------------- | ------------- | ----- | ----- | ------- |
> |                | SamSum        | SQuAD | IWSLT |         |
> | Our            | 48.09         | 90.59 | 38.62 | 2.17    |
> | Our+expo_decay | 48.01         | 90.56 | 38.59 | 2.25    |
>
>
> For the BERT model the results are:
>
> |       | SST-2 | QNLI | Speedup |
> | ---- | ----- | ---- | ------- |
> | Our            | 95.8  | 92.4 | 1.88    |
> | Our+expo_decay | 95.5  | 92.3 | 1.96    |
>
> We can observe that there is a slight accuracy drop, but there is some improvement in speedup, more likely due to a relaxed condition in the deeper exits as compared to the initial ones, and the percentage of samples exiting at the final layer decreases. These results can be replicated from our original source code.
>
> We thank you for your time and effort in reviewing our work, we hope that we clarified most of your doubts, if you have any more questions, please let us know, else we request you to please reassess the scores.

---

> > ### Comment · Reviewer_Cxuf · 2025-08-06
> > **Good work for NeurIPS conference**
> >
> > The authors addressed most of my concerns; therefore, I decided to raise my score to accept. I appreciate their comprehensive evaluation and ablation studies. Their method outperforms the depicted baselines, and the authors provide intuitive explanations for why this is the case.
> >
> > In my opinion, focusing on risk control is a reasonable approach in the context of early-exit architectures, as these networks rely on whether the confidence exceeds the threshold or not (a binary condition). Therefore, avoiding premature, incorrect, and overly confident predictions is a natural direction for research. It is worth mentioning that the authors specifically modify the training loss and reward functions (Equations 3 and 4) to achieve better integration with decision-making and to account for the computational cost of individual exits. These are key aspects of early-exit architectures.

---

> > > ### Author Response · Authors · 2025-08-06
> > > **Thanks for positive remarks!**
> > >
> > > Thank you sincerely for your encouraging feedback and for raising your score. We’re especially grateful that you highlighted the significance of our work and for considering it a candidate for inclusion in the conference.

---

> ### Comment · Area_Chair_UWUj · 2025-08-04
>
> Dear reviewer,
> Please engage with the authors' rebuttal, especially w.r.t. the additional empirical results.

---

### Official Review · Reviewer_hcfn · 2025-07-06

**Clarity:** 3
**Significance:** 3
**Originality:** 3
**Rating:** 5
**Confidence:** 4

**Summary:**

The paper presents a framework for selecting adaptive thresholds for early-exit neural networks. The motivation is that existing methods rely on a fixed threshold chosen from a static validation set, which can lead to incorrect predictions when the model is over-confident in the wrong class under distribution shift at inference. This reduces accuracy and undermines model reliability. To avoid this, the paper proposes to adaptively update a single global threshold online during inference using an upper-confidence-bound (UCB) multi-armed-bandit framework, with a probabilistic guarantee that the overall miscoverage risk stays within a tolerance, while improving computational efficiency by exiting at early layers. Since ground-truth labels are unavailable during inference, a neural network is trained offline to estimate the reliability of the prediction. This estimated reliability is used to select the best threshold by maximizing a reward function, which combines the product of confidence and reliability with an exit penalty that increases for later layers. Experimental results show that the proposed dynamic-threshold selection achieves significant speed-ups with minimal drops in performance.

**Questions:**

See weaknesses

**Ethical Concerns:**

["NO or VERY MINOR ethics concerns only"]

**Final Justification:**

This is a well-written paper that addresses a critical problem in the early-exit literature. The proposed solution is reasonable and represents a step towards improving the real-world practicality of such models.

**Limitations:**

Included but outside page limit

**Quality:**

3

**Strengths And Weaknesses:**

**Strengths**
* The motivation of the paper is clear and intuitive; the threshold for early exit decisions should be adaptive to account for distribution shift during inference.
* The formulation is interesting, using confidence and reliability of prediction in addition to penalizing late exits to improve prediction quality and computational efficiency simultaneously.
* Experimental results demonstrate the effectiveness of the approach on diverse tasks, where in some tasks the performance is better than the full model

**Weaknesses**
* The method still selects a global threshold which is adapted online based on previously observed samples. So if an OOD sample arrives, the exit decision is made based on the global threshold, and the model can still be overconfident and misclassify or exit at later depth. Instead, a per-sample based threshold would be more robust to abrupt distribution shifts.
* Because the reliability function g is trained offline, it is possible that it drifts under distribution shifts and the estimates are unreliable at inference time due to sub-optimal threshold selection. How can we be sure of the robustness of g under distribution shifts?

---

> ### Author Response · Authors · 2025-07-31
> **Rebuttal**
>
> Thanks for your positive and insightful remarks.
>
> **Q1:** The method still selects a global threshold which is adapted online based on previously observed samples. So if an OOD sample arrives, the exit decision is made based on the global threshold, and the model can still be overconfident and misclassify or exit at later depth. Instead, a per-sample based threshold would be more robust to abrupt distribution shifts.
>
> **A1:** We agree that the method uses a global threshold, and OOD samples may limit its adaptivity. That said, our method is more robust to OOD samples than standard existing  global-threshold methods. This is because in our case, the OOD samples are likely to be pushed to the last layer for inference, and this reduces the risk of misclassification from premature predictions. This robustness is due to the following reasons:
>
> 1) The learned threshold maximizes a reward function $C_i⋅(1−C_g)-\psi(i)$, where $C_i$​ reflects model confidence and $C_g$​ captures confidence reliability.
>
>
> 2) For OOD samples, both confidence $C_i$ and the reliability function $1-C_g$​ tend to be lower, as $C_i$ is the model confidence that is trained on a specific dataset distribution (training dataset) and as the distribution changes, the model confidence drops, similarly $C_g$ trained to identify high-loss regions in the training dataset distribution, responds with a high value depicting less reliability on that confidence (as model is likely to make higher loss for samples from different domain) and overall lowering the product $C_i⋅(1−C_g)$ which needs to surpass the threshold in our case (see Table 5). Hence OOD sample is likely to be deferred to the final exit, reducing the risk of misclassification due to premature prediction.
>
> Also, Prior work [1] has shown that early-exit models are reasonably robust to OOD data even with static thresholds. Our added reliability-aware gating further decreases the chance of exiting early on unreliable or OOD inputs.
> By per-sample adaptive threshold, we understand that you are asking for threshold to be chosen based on the sample itself. But this setup can increase complexity of the learning as the samples can be diverse and we have to deal with a large number of thresholds. This may affect the performance of EE models. One possibility is we can use contextual bandit setup where the choice of threshold depends on the observed context (per-sample), however this needs different model assumptions e.g., parameterized rewards.  Thanks for the suggestion, this can be explored as an independent work. We welcome any other ideas to model this problem better.
>
> [1] https://proceedings.mlr.press/v158/qendro21a/qendro21a.pdf.
>
> **Q2:** Because the reliability function g is trained offline, it is possible that it drifts under distribution shifts, and the estimates are unreliable at inference time due to sub-optimal threshold selection. How can we be sure of the robustness of g under distribution shifts?
>
> **A2:** We acknowledge that offline-trained functions may degrade under extreme or adversarial distribution shifts. However, our reliability predictor $g$ can handle realistic and gradual distributional shifts, such as those caused by slight domain changes (e.g., training on movie reviews and testing on book reviews) or mild corruptions (e.g., noisy or blurred images). Such shifts, though they affect the data distribution, do not drastically alter the confidence and loss landscape of the model, which $g$ relies on. This robustness is empirically validated in Tables 4 and 5 of our paper, where performance remains stable across various realistic distribution shifts.
>
>
> We thank you for your time and effort in reviewing our work, we hope that we clarified most of your doubts, if you have any more questions, please let us know, else we request you to please reassess the scores.

---

> ### Comment · Area_Chair_UWUj · 2025-08-04
>
> Dear reviewer, Please engage with the authors' rebuttal, especially w.r.t. the use of a global thresholds and the resulting OOD robustness.

---

> > ### Comment · Reviewer_hcfn · 2025-08-06
> >
> > The author's response addressess my questions, therefore I have increased my rating to accept the paper. The paper's formulation of the problem directly ensures that OOD robustness of EE models improves when using an adaptive threshold, compared to the fixed global threshold used by prior methods. Additionally, the method provides reliability estimates, both of which are very important in real-world scenarios.

---

> > > ### Author Response · Authors · 2025-08-07
> > > **Thank you for the positive remarks**
> > >
> > > We sincerely thank the reviewer for the positive remarks and for increasing the score. We are grateful for highlighting our key contribution to addressing the robustness of EE models through reliability estimates.

---

### Author Response · Authors · 2025-08-02
**Gentle reminder and a note**

Dear AC and reviewers,

We have worked hard for our rebuttal but got slightly delayed in posting the rebuttal due to some confusion on dates. We posted the rebuttal as an official comment. We request you to kindly consider our official comments as rebuttal and discuss on that. It will make our efforts fruitful.

Thanks in advance. Hoping to have a fruitful discussion phase.

---

### Note · Authors · 2025-08-12

Dear AC and Reviewers,

Thank you for the time and effort you have devoted to reviewing our paper and engaging in the discussion. We would like to take this opportunity to clearly restate and further emphasize the novelty, motivation, and significance of our contributions.

Early-exit (EE) neural networks offer a promising way to reduce inference time by exiting early when the model is sufficiently confident. However, a critical yet underexplored challenge remains: EE models can make highly-confident wrong predictions, especially when facing distribution shifts. This is not a minor issue; such predictions are concerning in reliability-sensitive applications, yet prior works have largely focused on improving speedup without adequately addressing this reliability gap.

Our work tackles this challenge by introducing a robust, adaptive, and theoretically grounded threshold-selection framework that operates during inference. We thanks Reviewer hcfn and Cxuf for acknowledging this. We have provided additional details to NGvD to clarify this.

The key ideas in our work are:

**1) Adaptation to distribution shifts:** Instead of fixing a threshold during training or tuning it offline, we let the model adaptively choose the best threshold at inference time. We setup this as a Multi-Armed Bandit (MAB) problem. Since threshold selection is data-driven, the method naturally adapts to changing input distributions without requiring retraining.

**2) Reward function designed for reliability:** The MAB is guided by a reward that accounts not only for raw model confidence but also for how reliable that confidence is, directly addressing the risk of high-confidence mistakes.

This formulation not only improves empirical performance under both in-distribution and out-of-distribution settings but also provides theoretical guarantees on the model’s risk. Importantly, these guarantees are not dependent on the training dataset (as they are in existing methods), making them robust to real-world deployment scenarios.

We believe that this combination of practical robustness, adaptive inference, and theoretical backing makes our work both timely and impactful. It opens the door for deploying early-exit models in critical applications from healthcare diagnostics to autonomous systems where reliability matters just as much as efficiency.

---

### Decision · Program_Chairs · 2025-09-17

**Decision:**

Accept (poster)

**Comment:**

This paper addresses the problem of risk control for early-exit neural networks (EENNs).  Whereas prior work has predominantly used confidence-based exiting (e.g. picking a confidence threshold on a validation set), this has the limitation of being non-robust to distribution shift.  This paper addresses this issue by formulating threshold-tuning as an online bandit problem, allowing the threshold to be calibrated for data that has undergone distribution shift.  Compelling results are shown on language and vision tasks.

The reviewers, in aggregate, had a favorable opinion of the work: 1 x reject, 2 x accept.  Reviewer NGvD criticized the work's novelty---specifically, the motivation for an online setting that necessitates calibrating an exit-threshold.  However, I concur with the other two reviewers, finding the work does indeed provide a novel approach to a perhaps niche but still practical setting of online calibration under data shift.  Yet, in subsequent drafts, the authors should work to better motivate the online setting.

While not mentioned by the reviewers, I see one issue with the draft that is in need of correction.  Section 2 claims "Rather than comparing early exits against the final layer’s predictions, we model the overall risk of the complete system, enabling our method to potentially surpass the performance of the base DNN itself, an important limitation overlooked by prior work".  This statement is incorrect: Jazbec et al. [30] do consider and support negative losses, which means that an early exit is performing better than the final exit.  See their Proposition 3 and surrounding discussion.  Please correct this claim in subsequent drafts.